# Myofibril and mitochondria morphogenesis are coordinated by a mechanical feedback mechanism in muscle

Jerome Avellaneda [1], Clement Rodier[1], Fabrice Daian [1], Nicolas Brouilly[1], Thomas Rival [1], Nuno Miguel Luis [1 ✉] & Frank Schnorrer [1 ✉]

Complex animals build specialised muscles to match specific biomechanical and energetic needs. Hence, composition and architecture of sarcomeres and mitochondria are muscle type specific. However, mechanisms coordinating mitochondria with sarcomere morphogenesis are elusive. Here we use *Drosophila* muscles to demonstrate that myofibril and mitochondria morphogenesis are intimately linked. In flight muscles, the muscle selector *spalt* instructs mitochondria to intercalate between myofibrils, which in turn mechanically constrain mitochondria into elongated shapes. Conversely in cross-striated leg muscles, mitochondria networks surround myofibril bundles, contacting myofibrils only with thin extensions. To investigate the mechanism causing these differences, we manipulated mitochondrial dynamics and found that increased mitochondrial fusion during myofibril assembly prevents mitochondrial intercalation in flight muscles. Strikingly, this causes the expression of cross-striated muscle specific sarcomeric proteins. Consequently, flight muscle myofibrils convert towards a partially cross-striated architecture. Together, these data suggest a biomechanical feedback mechanism downstream of *spalt* synchronizing mitochondria with myofibril morphogenesis.

[1] Aix Marseille University, CNRS, IBDM, Turing Center for Living Systems, Marseille, France. ✉email: nuno.luis@univ-amu.fr; frank.schnorrer@univ-amu.fr

Muscles power all voluntary animal movements. These movements are produced by arrays of myosin motors that are assembled together with titin and actin filaments into elaborate contractile machines called sarcomeres[1,2]. Hundreds of sarcomeres are connected into long chains called myofibrils that span the entire muscle fibre and thus mechanically connect two skeletal elements[3]. During muscle contraction each myosin motor head consumes one molecule of ATP per cross-bridge cycle to move myosin ~10 nm relative to actin and to produce a few piconewton of force[4]. Thus, sustained muscle contraction requires large amounts of ATP.

As ATP is most effectively produced by oxidative phosphorylation in mitochondria, muscles generally contain large amounts of mitochondria. However, mitochondrial content varies to a large extent between different muscle types and across species[5], suggesting that mitochondria biogenesis is adjusted to match the energetic requirements of muscle fibre types. A striking example are slow oxidative muscle fibres of mammals that are enduring muscles and thus strongly depend on high ATP levels. These fibres contain larger amounts of mitochondria compared to fast glycolytic fibres[6,7]. However, not only total mitochondrial content but also mitochondrial morphology is fibre-type dependent with more elongated mitochondria present in mammalian oxidative fibre types[6]. This suggests that mitochondria biogenesis is intimately linked to muscle fibre-type-specific physiology. However, the molecular mechanisms of this coordination are unclear.

Recent advances in high-resolution imaging revealed that mitochondrial morphologies in individual muscle fibres are not homogeneous. Mitochondria closer to the plasma membrane are generally more globular, whereas mitochondria in proximity to myofibrils are part of more complex networks[7]. Parts of the mitochondrial network contact the sarcomeric I-bands, other parts run in parallel to the fibre axis, in close proximity to the myofibrils[8]. Strikingly, the organisation of mitochondrial networks also depends on the muscle fibre type: oxidative fibres contain more mitochondria preferentially oriented in proximity to and in the direction of myofibrils, a phenomenon even more prominent in the heart, a muscle that strictly depends on ATP production by oxidative phosphorylation[9]. Hence, ATP production is located close to the ATP consuming contractile motors. However, little is known about the mechanisms of how myofibril and mitochondria development are coordinated to match the energetic requirements with the contractile properties of muscle fibres.

To investigate the interplay between myofibrils and mitochondria, we turned to *Drosophila* and compared two different *Drosophila* muscle types, indirect flight muscles and leg muscles. Indirect flight muscles of insects are specialised to combine high power output with endurance and thus use oxidative metabolism. *Drosophila* flight muscles oscillate at 200 Hz and produce up to 80 Watt power per kg of muscle mass during long flight periods[10–12]. Hence, the ATP demand of these muscle fibres during flight is very high. The fast oscillations are triggered via a stretch-activation mechanism, which is achieved by a specialised architecture of the contractile myofibrils, called fibrillar morphology, with individualised myofibrils that are not laterally aligned with their neighbours[13]. With its strict aerobic metabolism and its stretch-activation mechanism requiring high mechanical tension, insect flight muscles biomechanically and energetically resemble the mammalian heart muscle[14,15].

In contrast, the other adult *Drosophila* body muscles found in legs or abdomen show a regular cross-striated myofibril morphology with neighbouring myofibrils aligned laterally, resembling mammalian skeletal muscle fibres architecture[16,17]. They use a normal synchronous contraction mechanism. Thus, their energy requirements are strikingly different from flight muscles.

Here, we compared myofibril and mitochondria morphologies between indirect flight and leg muscles of *Drosophila* and found that flight muscle mitochondria are mechanically squeezed against myofibrils maximising their contact areas and isolating neighbouring myofibrils. We discovered that mitochondrial intercalation between myofibrils coincides with myofibril assembly. Strikingly, if intercalation is prevented by increased mitochondrial fusion, fibrillar flight muscles express sarcomeric proteins specific to the cross-striated leg muscle type resulting in a partial conversion to cross-striated fibre morphology. This suggests a mechanical interplay between mitochondria dynamics and myofibril development, which triggers a feedback mechanism coordinating mitochondria with myofibril morphogenesis.

## Results

**Muscle type-specific mitochondria morphology is instructed by Spalt.** In order to examine the regulation of mitochondria biogenesis and myofibril morphogenesis in different fibre types, we chose *Drosophila* adult indirect flight muscles and leg muscles as models. Flight muscles consist of dorso-ventral muscles (DVMs) and dorso-longitudinal muscles (DLMs)[18]. As both show a very similar morphology, we focus on the DLMs and for simplicity call them flight muscles in the remainder of the manuscript. We visualised myofibril morphology with phalloidin and mitochondria morphology by expressing GFP fused to a mitochondrial matrix targeting signal (mito-GFP) with *Mef2*-GAL4. Flight muscles show the expected fibrillar myofibril morphology with individualised myofibrils (Fig. 1a, b′)[17]. Flight muscle mitochondria are densely packed around the individual myofibrils, adopting an elongated shape along the myofibril axis, consequently physically isolating neighbouring myofibrils (Fig. 1b, b″).

In contrast, leg muscles have cross-striated myofibrils, which align laterally to form a tube, whose centre is devoid of myofibrils (Fig. 1a, c′–f′, i)[17]. Interestingly, leg muscle mitochondria do not intercalate within the cross-striated myofibrils, but are present both peripherally (Fig. 1c) and centrally in the tube, where they are strongly concentrated (Fig. 1f, k). They appear to be largely excluded from the area occupied by the cross-striated myofibrils, with only small mitochondrial extensions contacting the sarcomeric I-bands (Fig. 1c–e). Such a specific mitochondrial-myofibril contact area is not found in the fibrillar flight muscles, however, the overall mitochondrial content, when normalised to the actin content of flight and leg muscles is comparable (Fig. 1j).

It was shown previously that the formation of fibrillar flight muscle requires the zinc-finger transcription factor Spalt (Spalt major, Salm)[17], however effects on mitochondria morphology had not been explored. Interestingly, we found that knock-down of *spalt* in flight muscles during development using *Mef2*-GAL4 not only transforms myofibrils into a cross-striated tubular morphology (Fig. 1g–i and Supplementary Fig. 1) but also converts the simpler mitochondrial morphology of flight muscles into a leg-specific type with centrally concentrated mitochondria, which contact the sarcomeric I-bands with thin extensions (Fig. 1h–h″). Taken together, these data show that the physiologically and mechanically distinct muscle fibre types of adult flies display strikingly different mitochondrial morphologies. In flight muscles, mitochondria and myofibril morphologies are both instructed by the transcriptional regulator Spalt.

**Flight muscle mitochondria elongate in proximity to myofibrils.** In order to examine in more detail mitochondria morphology in relation to myofibril structure, we developed a method to better quantify mitochondria morphologies in the different muscle types. To be able to delineate mitochondrial shape in an automated way in flight muscles, we established a live dissection

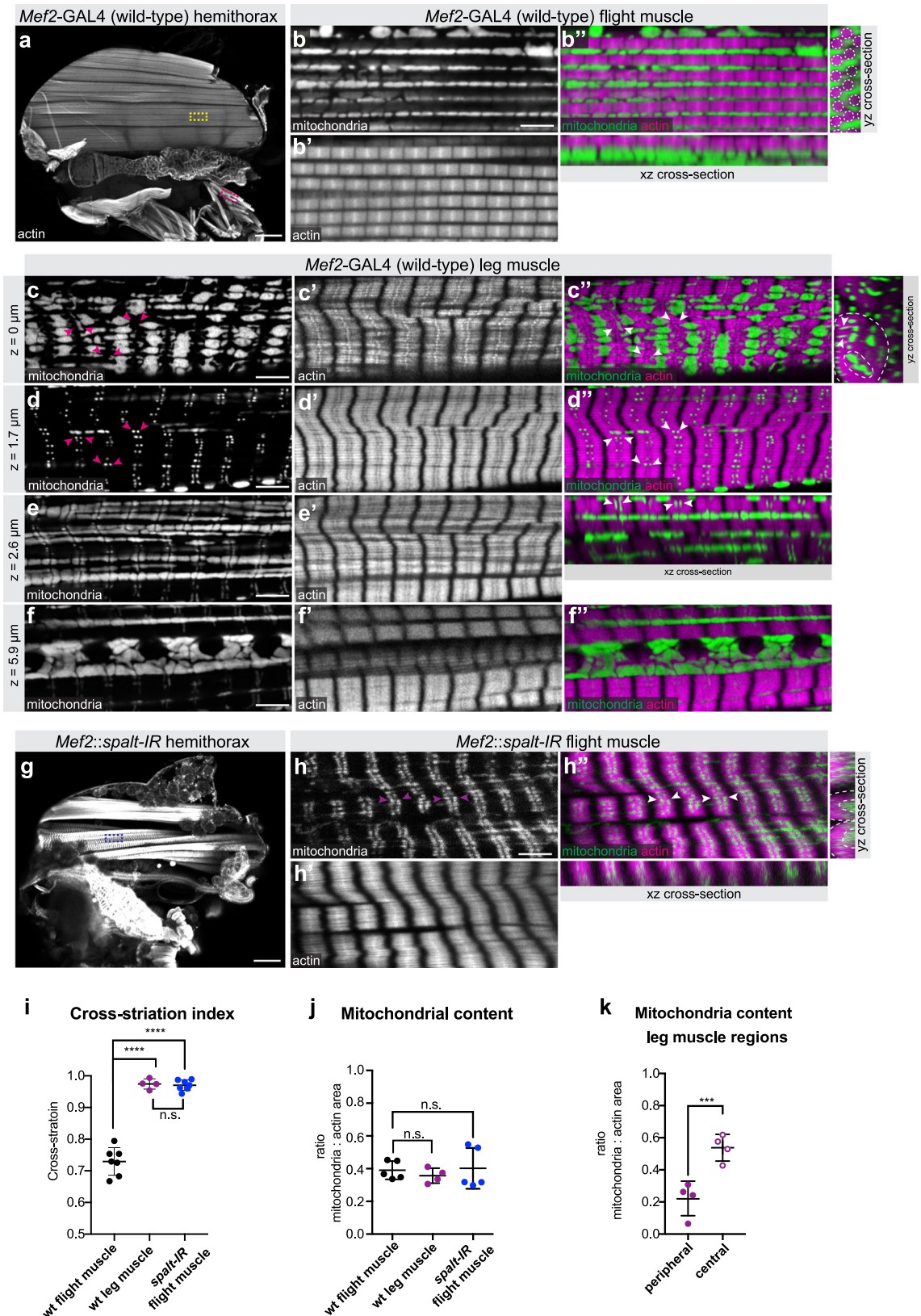

method avoiding fixation. Additionally, we generated a marker line labelling the mitochondrial outer membrane by fusing GFP to the mitochondrial outer membrane localisation signal of Tom20, here named MOM-GFP (see Methods section). When expressed in flight muscles with *Mef2*-GAL4, MOM-GFP delineates the mitochondrial outer membrane (Fig. 2a), which

enabled us to segment and reconstruct individual mitochondria in three dimensions using a deep learning network (Fig. 2b, c, Supplementary Fig. 2a–f and Supplementary Movie 1). These data show that the average volume of flight muscle mitochondria is ~3–4 μm³ (Fig. 2d). Most mitochondria adopt a simple elongated ellipsoid-like shape with the long axis of the ellipsoid oriented in

**Fig. 1** *spalt* **regulates muscle type-specific mitochondria morphogenesis. a–f** *Mef2*-GAL4 (wild type) hemithorax (**a**), flight muscle (**b**) and leg muscle (**c–f**) stained with phalloidin to visualise actin (magenta) and expressing mito-GFP to visualise the mitochondrial matrix (green). Yellow and magenta boxes in **a** indicate representative regions of flight and leg muscles magnified in **b–f**. Single confocal plane as well as and *xz yz* cross-sections are shown (**b″**). Note the individualised myofibrils (dotted circles) surrounded by densely packed mitochondria. **c–f** Leg muscle top (**c**), middle (**d, e**) and central slice (**f**) showing the tubular fibre morphology (*yz* cross-section), cross-striated myofibrils and complex mitochondrial shapes filling the surface and the centre of the myofiber and contacting the sarcomeric I-bands with thin extensions (magenta and white arrow heads). **g, h** *Mef2::spalt-IR* hemithorax (**g**) and flight muscle (**h**) display tubular fibre morphology (**h″** *yz* cross-section), cross-striated myofibrils and centrally located mitochondria with thin extension towards the I-bands (arrow heads). **i–k** Quantification of the lateral fibrillar alignment called cross-striation index in muscle (**i**; $n = 6, 4, 6$ animals respectively see Supplementary Fig. 1), the relative mitochondria content (**j**, relative to myofibril content; $n = 5, 4, 5$ animals respectively) and the mitochondria content in leg muscle regions (**k**; $n = 4$). Dotted lines on the *yz* cross-sections of **c″** and **h″** represent the regions measured. Note that higher mitochondria density in the centre of leg muscles. In all plots, individual circles represent individual animals, for each a minimum of five measurements was done, and mean ± standard-deviation (SD) is indicated. Significance from two-tailed unpaired *t*-tests is denoted as *p*-values ***$p \leq 0.001$ or ****$p \leq 0.0001$. (n.s.) non-significant. Scale bars are 100 μm (**a, g**) and 5 μm (**b, c–f, h**).

the direction of the myofibrils (Fig. 2c, e and Supplementary Fig. 2d). Specific extensions towards the myofibrils are absent, instead very large contact areas between myofibrils and mitochondria are likely present in flight muscle.

To resolve these contact areas in more detail we applied serial block-face electron microscopy and indeed could verify the intimate contacts with virtually no detectable space between mitochondria and myofibrils (Fig. 2f and Supplementary Movie 2). By reconstructing myofibrils and mitochondria in three dimensions we found that the majority of elongated mitochondria are squeezed against individual myofibrils resulting in round indentations in the mitochondria that cover about half of a myofibril circumference (Fig. 2g, h and Supplementary Movie 2). Mitochondria do not form networks but are rather individualised with an average volume of an individual mitochondrion of 3.9 μm³, which is in good accordance with our light microscopy quantifications. These data demonstrate that myofibril and mitochondria morphologies are intimately linked in flight muscles and thus suggest that myofibril development is highly coordinated with mitochondria morphogenesis.

**Leg muscle mitochondria acquire complex shapes.** To segment the complex shapes of leg muscle mitochondria we used the mitochondrial matrix marker mito-GFP expressed with *Mef2*-GAL4 (Fig. 2i, j and Supplementary Movie 3). This allowed us to attempt a 3D reconstruction of leg muscle mitochondria (Fig. 2k and Supplementary Movie 4). However, the success of the automated segmentation of individual mitochondria was limited by their complex shapes and thin extensions. Manual reconstruction displayed these complex shapes with elongated structures extending in 3D, and particularly prominent extensions towards the sarcomeric I-bands (Supplementary Fig. 2g–h″ and Supplementary Movies 5 and 6). Thus, leg muscle mitochondria display more complex shapes compared to flight muscle mitochondria.

As light microscopy resolution does not allow to assess connectivity of the mitochondrial network in detail, we have performed serial block-face electron microscopy of a representative leg muscle located in the coxa of the second thoracic segment (Fig. 2l and Supplementary Fig. 2i). A 3D segmentation revealed the detailed shapes of the individual leg muscle mitochondria, confirming that they form rods or ellipses largely located above or below the cross-striated myofibrils, with their longest axis present in the orientation of the myofibrils (Fig. 2m–o and Supplementary Fig. 2j, k). Additionally, the detailed 3D reconstruction allowed us to visualise the many individual thin mitochondria extensions protruding towards the sarcomeric I-bands, flanking the Z-discs, a unique feature of leg muscle mitochondria (Fig. 2m, p, q and Supplementary Movie 7). However, in most cases these thin extensions appear not to be connected to neighbouring mitochondria allowing us to calculate an average volume of

<1 μm³ per mitochondrion (Fig. 2n). Thus, leg muscle mitochondria organise into rather complex shapes above and below the aligned cross-striated myofibrils, and hence are strikingly different from flight muscle mitochondria.

**Flight muscle mitochondria morphology is ruled by mechanical pressure.** We found that flight muscle mitochondria are in intimate contact with myofibrils and acquire an elongated shape. We have shown in the past that myofibrils are under significant mechanical tension during development and that this tension is required to build linear myofibrils[19,20]. Hence, we hypothesised that myofibril tension creates a pushing force against mitochondria that constrains them into the observed ellipsoid shape. To test this hypothesis, we applied our live dissection protocol of flight muscles combining a marker for myofibrils (*UAS-Cherry-Gma*) with live mitochondria markers. Live dissection occasionally resulted in regions where myofibrils were mechanically severed (Fig. 3a). As shown above, areas with intact parallel myofibrils show elongated ellipsoid-shaped mitochondria with their long axis oriented in the direction of the myofibrils (Fig. 3b, d). Strikingly, severing myofibrils results in a dramatic rounding up of all neighbouring mitochondria into spheres (Fig. 3c, e). This transition was observed with both mitochondrial markers, MOM-GFP (Fig. 3c) as well as with mito-GFP (Fig. 3e′). Interestingly, no obvious connection between the rounded mitochondria and the myofibrils remained visible within the severed area (Fig. 3e″) strongly suggesting that mechanical pressure created by the tense myofibrils, rather than specific protein–protein binding, pushes mitochondria into their elongated shape covering the myofibrils.

To explore the mechanical impact from myofibrils on mitochondrial shape further we combined our live dissection protocol with a flight muscle-specific allele of myosin heavy chain (*Mhc[10]*), which specifically disrupts flight muscle myofibrils[21,22]. Remarkably, interfering with myofibril development and hence reducing the mechanical constraint on mitochondria results in a dramatic change of mitochondrial morphology with mitochondria acquiring spherical shapes in *Mhc[10]* mutant flight muscles (Fig. 3f–g″). A similar observation had already been documented in the original electron microscopy images of *Mhc[10]* mutant flight muscles[23]. Together, these data demonstrate a tight mechanical interaction between the myofibrils and the mitochondria with the tension exerted from the myofibrils squeezing mitochondria into elongated shapes.

**Mitochondrial dynamics impacts myofibril development.** Mechanical shaping of mitochondria by myofibrils should require a close contact between the two during muscle development. Mitochondria are highly dynamic organelles whose morphologies are defined by a delicate balance between mitochondrial fusion

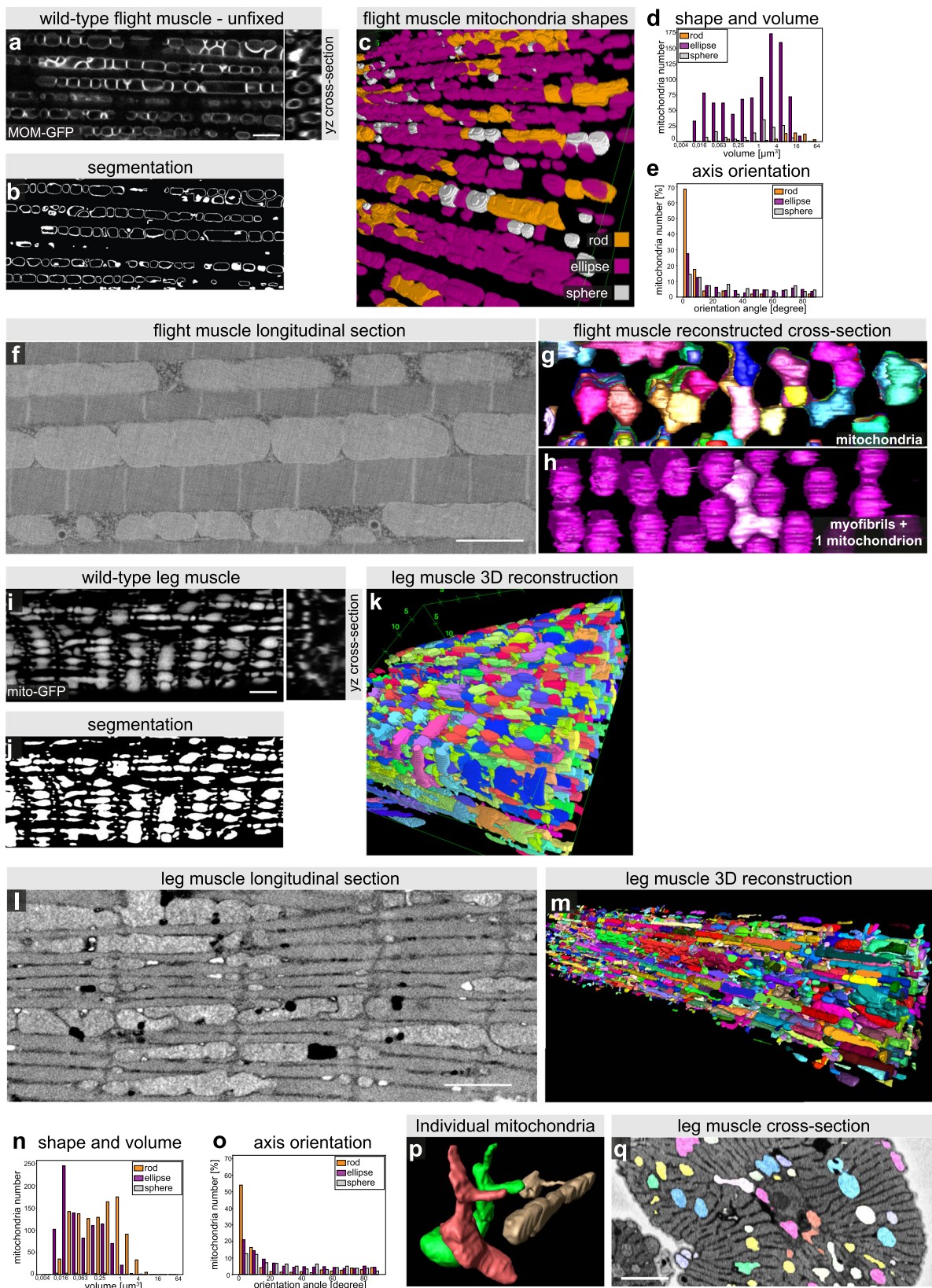

and fission[24–26]. Thus, we hypothesised that changing fusion or fission rates may not only change mitochondrial shapes but also impact on myofibril development. To test our hypothesis, we reduced mitochondrial fusion by knocking down *Mitochondrial associated regulatory factor* (*Marf*), a mitofusin required for outer mitochondrial membrane fusion in flight muscles[27–29], with

*Mef2*-GAL4 in muscles (*Mef2::Marf-IR*). Flight muscle fibre morphology of *Mef2::Marf-IR* flies is normal, however flight function, as assayed by a flight test, is impaired (Fig. 4a–c). As to be expected, reduced fusion rate results in smaller mitochondria in *Mef2::Marf-IR* flight muscles, which adopt a spherical instead of an ellipsoid shape (Fig. 4d′, e′, k). However, these small

**Fig. 2 Quantification of mitochondrial morphology in muscle types. a–e** Highly resolved confocal sections of unfixed alive flight muscle mitochondria labelled with MOM-GFP expressed with *Mef2*-GAL4 (**a**) to segment the mitochondria outlines using machine learning (**b**, see Supplementary Fig. 2). In all, 3D segmentation of individual flight muscle mitochondria using Fiji with classification of individual mitochondria based on shape classifiers (**c**), see Methods section for the classification parameters. Total mitochondria number and their volumes in a 67.5 μm × 67.5 μm × 6.7 μm volume (**d**). Note the preferred orientation of the long mitochondrial axis with the axis of the myofibrils (**e**). **f–h** Serial block-face electron microscopy of adult flight muscles, showing a longitudinal view (**f**). Note the intimate contact of mitochondria and myofibrils. Cross-section view of a 3D reconstruction of individual mitochondria shown in different colours (**g**) and of the myofibrils in magenta with one mitochondrion in light pink (**h**, Supplementary Movie 2). Note the mitochondrial indentations caused by pushing myofibrils. **i–k** Fixed leg muscle mitochondria labelled with mitochondrial matrix GFP (mito-GFP) expressed with *Mef2*-GAL4 (Supplementary Movie 3). A representative peripheral (top) section of the z-stack (also used in Fig. 1c) and a yz-cross-section orthogonal view are shown (**i**). Interactive Watershed using Fiji allowed segmentation (**j**) and 3D reconstruction of individual mitochondria (**k**, Supplementary Movie 4). **l–q** Serial block-face electron microscopy of adult a coxa muscle from a second thorax segment leg showing a longitudinal view (**l**). Note the small mitochondria parts located next to the I-bands, which extend from larger mitochondria seen in the 3D reconstruction (**m**, Supplementary Movie 7). **n, q** Mitochondria were individually segmented, allowing to measure total mitochondria number and their volumes in a 56 μm × 12.9 μm × 8.73 μm volume based on shape classifiers (**n**), see Methods section for the classification parameters. Note the orientation of the long mitochondrial axis with the axis of the myofibrils (**o**), similar to flight muscle mitochondria, despite the perpendicular extensions visible in individual mitochondria (**p**), yz-cross-section orthogonal view (**q**). Scale bars are 5 μm in **a**, **i** and 2 μm in **f**, **l**, **q**.

mitochondria intercalate normally between myofibrils resulting in wild-type shaped individualised fibrillar myofibrils with normal myofibril diameter and normal sarcomere length (Fig. 4d, e and Supplementary Fig. 3e, f). Also, the mitochondrial content is comparable to wild type (Fig. 4j). We observed the same phenotype when we increased mitochondrial fission rate during development by overexpressing *Dynamin related protein 1* (*Drp1*), a regulator of outer mitochondrial membrane fusion[30] with *Mef2*-GAL4 (*Mef2::Drp1*) (Supplementary Fig. 3a–f). These data suggest that smaller mitochondria cannot sustain flight but are compatible with intercalating between myofibrils and thus enable normal fibrillar myofibril development of flight muscles.

In an attempt to convert flight muscle mitochondria into larger networks, we performed the converse experiment and increased mitochondrial fusion by overexpressing *Marf* during development using two differently strong UAS-*Marf* constructs (*Mef2::Marf-1* and *Mef2::Marf-2*)[27,28]. In both cases, over-expression of *Marf* with *Mef2*-GAL4 results in fewer flight muscles, likely due to reduced growth during development (Fig. 4f, g). This was not caused by a block in myoblast fusion, as myoblast fusion is normal during development of *Mef2::Marf-1* muscles (Supplementary Fig. 4). Interestingly, remaining flight muscle fibres show a dramatic change in their myofibril organisation with neighbouring myofibrils aligning laterally along the Z-discs and M-bands, mimicking the cross-striated leg muscle morphology (Fig. 4h, i, l). The total mitochondrial content is similar to control flight muscles (Fig. 4j), however *Marf* over-expression results in an exclusion of mitochondria from the myofibril layer, similar to our observations in leg muscles (Fig. 4h″, i″). In some cases, in particular with the stronger *Marf-1* construct, perfect tubular muscles are generated with all myofibrils lining the outside of a tube and mitochondria located centrally (Supplementary Fig. 3g–g″). This transformation from fibrillar to cross-striated myofibril morphology was also observed when mitochondria fission was suppressed by expression of dominant negative *Drp1* (*Mef2::Drp1-k38a*)[29,31] (Supplementary Fig. 3h–k) and thus is not a specific effect of *Marf* over-expression but generally caused by tipping mitochondrial dynamics towards more fusion. Taken together, these results imply that increasing the mitochondrial fusion rate impacts myofibril development such that individual fibrillar myofibrils cannot form and instead fuse together to form cross-striated myofibrils.

**Mitochondrial fusion shifts transcription towards cross-striated fate**. To decipher the mechanism of how a change in mitochondrial dynamics can impact myofibril morphology we investigated the expression of sarcomeric protein isoforms that

are specifically expressed in fibrillar flight or cross-striated leg muscle types[32]. We used GFP fusions of large genomic fosmid clones that recapitulate the endogenous expression dynamics with the added advantage to allow the quantification of expression levels without the need of antibodies[33]. Interestingly, we found that levels of both the flight muscle specific actin isoform Act88F-GFP and the flight muscle specific myosin binding protein Flightin (Fln-GFP) are strongly reduced in flight muscles overexpressing *Marf* (*Mef2::Marf-1*; Fig. 5a–h). Conversely, Kettin (Kettin-GFP), a short isoform of the *Drosophila* titin homologue Sallimus, which is expressed at high levels in wild-type leg muscles[16], is boosted in *Mef2::Marf-1* flight muscles (Fig. 5i–l). Hence, an increase in mitochondrial fusion during muscle development results in a transcriptional shift towards a more cross-striated muscle fibre-type fate, which may contribute to the observed cross-striation phenotype.

A simple explanation of the observed phenotype would be that over-expression of *Marf* interferes with flight muscle fate patterning at an early stage of development. To investigate this, we quantified the expression levels of the Zn-finger transcriptional regulator Spalt, which was shown to be required and sufficient to induce fibrillar muscle fate[17]. Spalt is expressed at high levels immediately after myoblast fusion in the flight muscle myotubes[17]. Thus, we quantified Spalt protein expression during early flight muscle development at 24 h after puparium formation (APF) and found that Spalt levels in *Mef2::Marf-1* myotubes are comparable to wild type (Fig. 5m–o). Furthermore, we investigated an early Spalt target, the alternative splicing regulator Bruno (Aret), which is specifically expressed in developing flight muscles downstream of Spalt[16]. Bruno levels in *Mef2::Marf-1* myotubes are also comparable to wild type (Fig. 5p–r). This strongly suggests that flight muscle fate is induced normally in *Mef2::Marf-1* myotubes and, as consequence, that increased mitochondrial fusion impacts flight muscle development downstream of Spalt.

**Developmental timing of mitochondrial dynamics has differential impact on myofibril development**. To better define the stage at which increased mitochondrial fusion can impact myofibril development, we tested a series of different GAL4-driver lines that are active at different stages of flight muscle development, in comparison to *Mef2*-GAL4 which is continuously active during all stages[34]. When *Marf* is over-expressed using *him*-GAL4 or *1151*-GAL4, restricting the over-expression to myoblasts and early stages of myoblast fusion, ending shortly after 24 h APF[19], myofibrils and mitochondria show a wild-type morphology in adult flight muscle and flies can fly (Fig. 6a–e). This

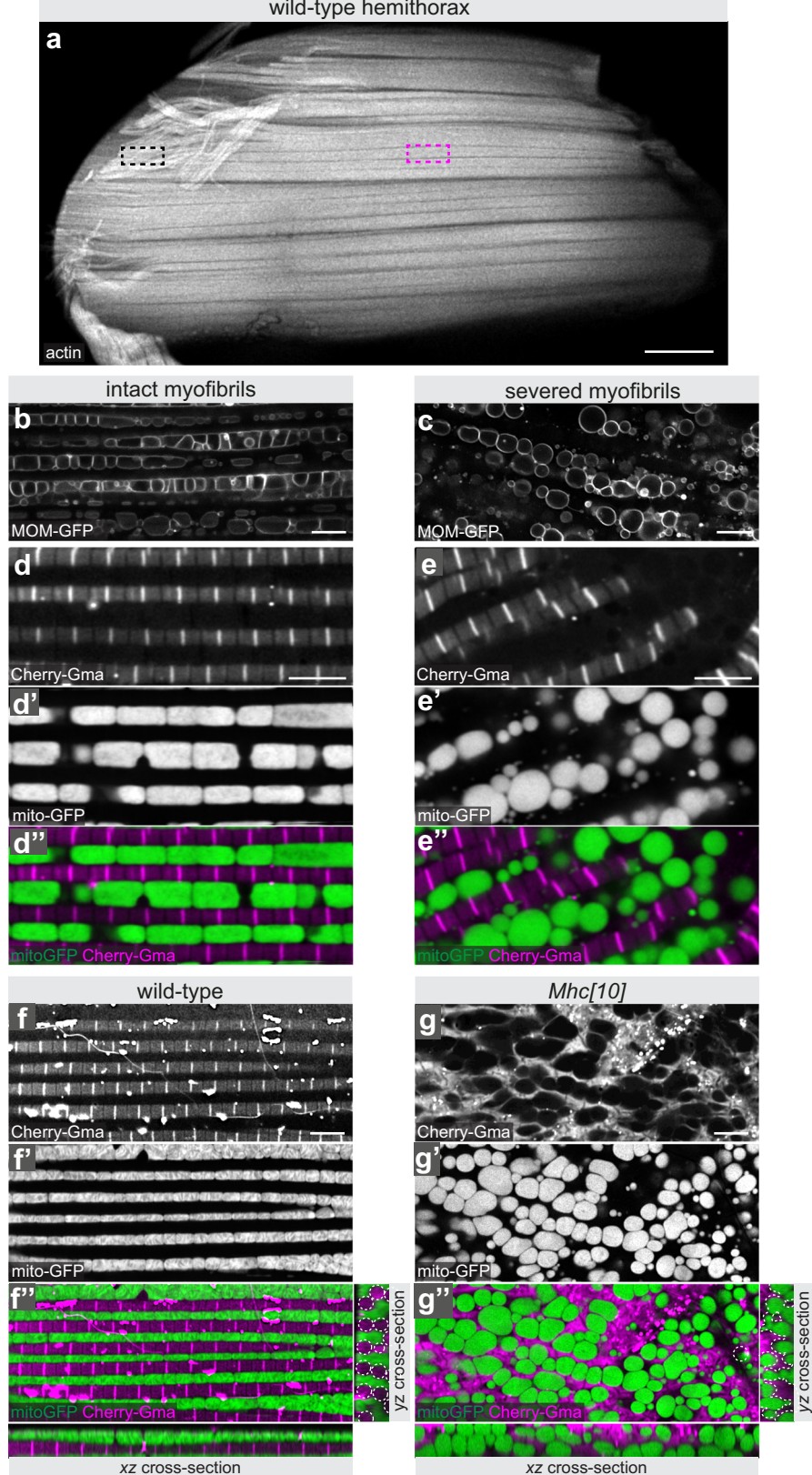

**Fig. 3 Myofibrils mechanically shape flight muscle mitochondria. a** Live dissection of hemithorax in which actin has been labelled with Cherry-Gma expressed with *Mef2*-GAL4. Black rectangle indicates a severed area, in which myofibrils have been mechanically cut, magenta rectangle marks an intact area. **b**–**e** High magnification confocal sections of intact (**b**, **d**) and severed areas (**c**, **e**) of unfixed flight muscles in which mitochondria have been labelled with MOM-GFP (expressed with *Mef2*-GAL4) (**b**, **c**) or with mito-GFP together with Cherry-Gma to label myofibrils (**d**, **e**). Note the spherical mitochondria shape and their disengagement from the myofibrils in the severed areas. **f**, **g** High magnification of unfixed flight muscle from wild type (**f**) or *Mhc[10]* mutant (**g**) adults genetically labelled with Cherry-Gma to label myofibrils (expressed with *Mef2*-GAL4) (**f**, **g**) and mito-GFP to label mitochondria (**f′**, **g′**). Note how similar the rounded *Mhc[10]* mitochondria (**g′**) are to the ones in severed myofibrils in **e′**. Scale bars are 100 μm (**a**) and 5 μm (**b**–**g**).

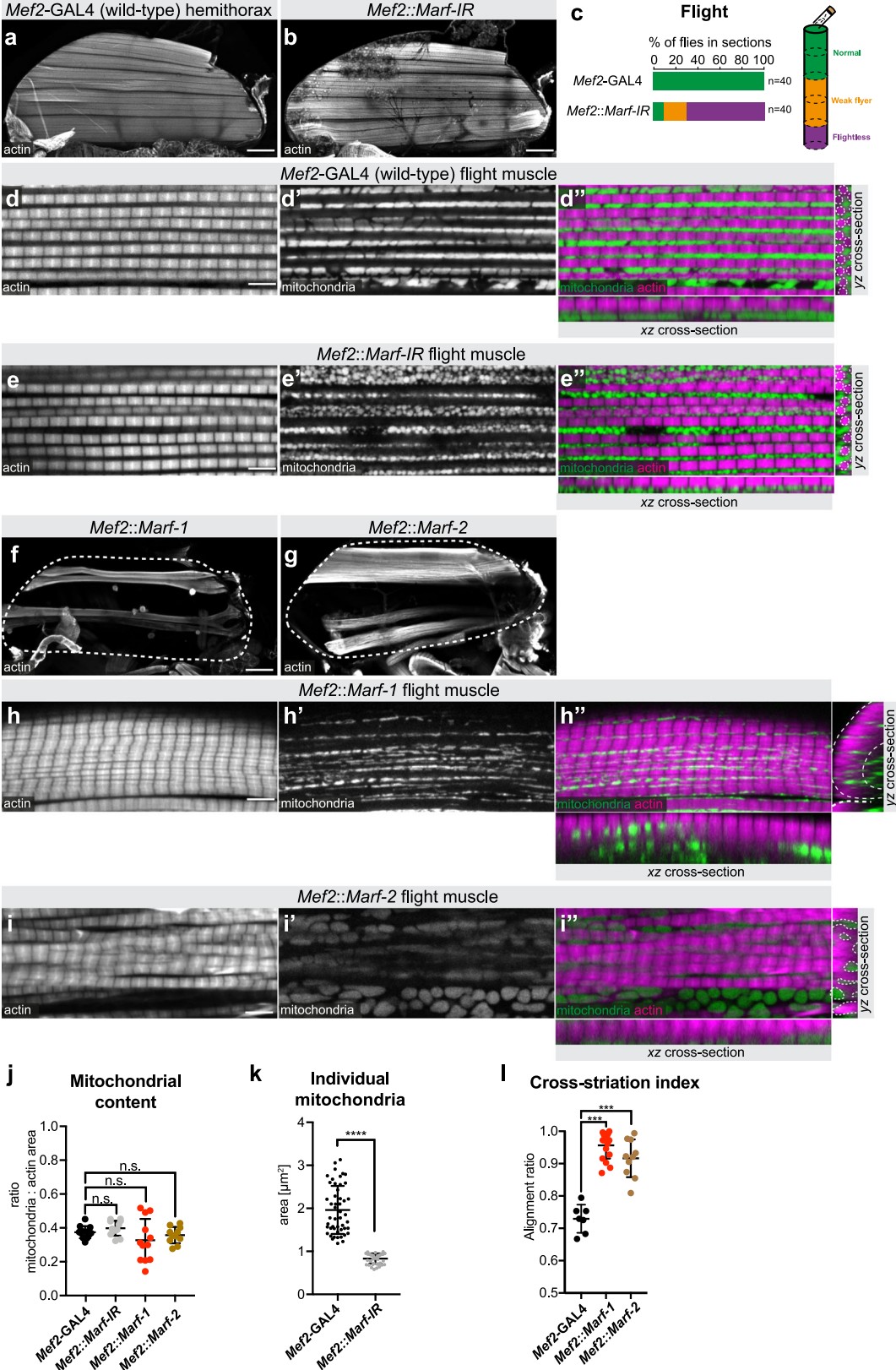

suggests that increasing mitochondrial fusion during myoblast and early myotube development, before myofibrils start to assemble, does neither impact myofibril development nor flight muscle function.

To over-express *Marf* during myofibril assembly we chose *Act88F*-GAL4, which is specifically and strongly expressed in flight muscles starting at about 24 h APF[35] and thus includes all stages of myofibril assembly and myofibril maturation[36]. Strikingly, *Act88F::Marf-1* flight muscles show a severe myofibril phenotype, ranging from enlarged myofibril diameter (Fig. 6g, j, n) to partially cross-striated myofibrils (Fig. 6h, k, m). Although total mitochondrial content is similar to wild type (Fig. 6l), mitochondria are

**Fig. 4 Mitochondria dynamics can impact myofibril development. a–i** Adult hemi-thoraces (**a**, **b**, **f**, **g**), flight muscles (**d**, **e**, **h**, **i**) and flight test (**c**) of the indicated genotypes. Actin has been visualised with phalloidin, mitochondria with mito-GFP. White dashed lines outline the myofibrils in the yz cross-sections (**d″**, **e″**). Note the small round mitochondria present between normal myofibrils upon *Marf* knock-down (**e**). *Marf* over-expression using *UAS-Marf-1* or *UAS-Marf-2* with *Mef2*-GAL4 causes fibre atrophy (**f**, **g**) and cross-striated myofibrils (**h**, **i**). Note that the mitochondria are largely separated from the aligned myofibrils, outlined by the dashed white line on the yz cross-sections (**h″**, **i″**). **j–l** Quantification of mitochondrial content (relative to actin area) (**j**; n = 12, 12, 12, 11 animals, respectively), of individual mitochondrial area in a single confocal section (**k**; n = 6894 and 14443 total mitochondria, from five animals in each case) and the cross-striation index of the indicated genotypes (**l**; n = 6, 17, 10 animals, respectively). In all plots the mean ± standard-deviation (SD) is indicated, each dot the value from single animals (**j**, **l**) or confocal sections (**k**), and significance from two-tailed unpaired *t*-tests is denoted as *p*-values ***$p \leq 0.001$, ****$p \leq 0.0001$. n.s. non-significant. Scale bars are 100 μm (**a**, **b**, **f**, **g**) and 5 μm (**d**, **e**, **h**, **i**).

often excluded from the inter-myofibril space (Fig. 6j″, k″), thus creating space for myofibril diameter growth or myofibril alignment towards a cross-striated phenotype. As a consequence, *Act88F::Marf-1* flight muscles cannot support flight (Fig. 6e). Similar to the continuous over-expression of *Marf* with *Mef2*-GAL4 we found that the *Act88F::Marf-1* flight muscles also show a strong reduction in the flight muscle specific proteins Act88F-GFP and Fln-GFP (Supplementary Fig. 5a–h) as well as a mild gain of the leg muscle-specific Kettin-GFP (Supplementary Fig. 5i–l). We conclude from these results that induction of mitochondria hyper-fusion specifically during stages of myofibril assembly and myofibril maturation strongly impacts myofibril growth and spacing as well as flight muscle-specific protein expression.

This interpretation is consistent with the normal temporal expression dynamics of fusion promoting *Marf* and fission promoting *Drp1* during flight muscle development[36]. At RNA level, *Marf* expression is boosted only from 30 h APF; in contrast, *Drp1* expression is down-regulated only after 48 h APF. Thus, the balance between these two factors changes only after 48 h APF strongly favouring mitochondria fusion (Supplementary Fig. 6). This might be one mechanism to restrict mitochondrial growth to stages after myofibril assembly[37,38].

**Mitochondria intercalate during myofibril assembly.** As we know little about the interplay between mitochondria and myofibrils during myofibril assembly and maturation stages, we wanted to explore this further. We dissected wild-type flight muscles at 24 h APF, a stage at which a dense network of actin filaments is present, while myofibrils are still undefined. We found that mitochondria form a widespread filamentous network of tubules that is largely separated from the dense actin filament mesh at 24 h APF (Fig. 7a, b and Supplementary Movie 8). When myofibrils have just assembled at 32 h APF, the mitochondria network has redistributed and mitochondria have intercalated between the myofibrils. As a consequence, myofibrils are indivi-dualised and are not in physical contact with neighbouring myofibrils (Fig. 7c, d and Supplementary Movie 8). This finding is supported by electron microscopy data that found mitochondria present between assembled myofibrils at 32 h APF[22]. Together, these data show that mitochondria and myofibrils are in close proximity directly after myofibrils have been assembled, sug-gesting a potential role of mitochondrial intercalation for fibrillar flight muscle morphogenesis.

**Fine-tuned mitochondria dynamics enables mitochondrial intercalation.** To test the functional relevance of mitochondrial intercalation we explored myofibril and mitochondrial morphologies after overexpressing *Marf* with *Mef2*-GAL4. We found that continuous mitochondrial hyper-fusion in flight muscles results in clustering of most mitochondria in a few areas outside of the actin filament mesh at 24 h APF (Fig. 7e, f and Supplementary Movie 8). Interestingly, these mitochondrial net-works are also maintained at 32 h APF preventing clustered

mitochondria to intercalate between the forming myofibrils in *Mef2::Marf-1* flight muscles (Fig. 7g, h and Supplementary Movie 8).

To investigate the differences of mitochondria networks at the ultrastructural level we have performed electron microscopy of flight muscles at 24 h APF. Consistently with the light microscopy data, we found that mitochondria are individualised and acquire an elongated tubular morphology in wild type. In contrast, *Marf* over-expression results in the formation of large mitochondrial clusters of abnormal shape at 24 h APF (Fig. 7i–l). This is consistent with a block of mitochondrial intercalation at 24 h APF in *Mef2::Marf-1* flight muscles.

As muscle development often appeared compromised when *Marf* was strongly over-expressed with *Mef2*-GAL4, we wanted to explore the developmental phenotype in more detail using the later *Act88F*-GAL4 driver line. In *Act88F::Marf-1* flight muscles myofibrils assemble well at 32 h APF, but as in *Mef2::Marf-1*, most mitochondria clump together in large networks that are physically separated from the myofibril layer (Fig. 8a–d and Supplementary Movie 9). Hence, mitochondrial intercalation is also strongly compromised in *Act88F::Marf-1* 32 h APF flight muscles.

Assembled myofibrils mature from 32 h to 48 h APF and display very regular sarcomeric patterns at 48 h APF[36] (Fig. 8e, f and Supplementary Movie 9). We wanted to investigate if the mitochondrial intercalation defect is maintained during myofibril maturation and how this impacts myofibril development. Indeed, we often found that mitochondria of 48 h APF *Act88F::Marf1* flight muscles stay networked in large clusters (Fig. 8g, h and Supplementary Movie 9). These mitochondria clusters are sometimes even present in the centre of a tube formed by closely aligned myofibrils (Fig. 8g″). Thus, mitochondria physically separate the maturing myofibrils at 48 h APF in wild-type flight muscles, whereas hyper-fused mitochondria networks fail to do so. This provides a mechanistic explanation, why the intercalation block can result in myofibril diameter overgrow and often in lateral alignment of neighbouring myofibrils resulting in cross-striated fibres at the adult stage (see Fig. 9). Thus, balanced mitochondrial dynamics enables mitochondria to physically isolate the maturing myofibrils to support fibrillar flight muscle development.

## Discussion

Here we are proposing that mitochondria and myofibril mor-phogenesis are coordinated by a mechanical feedback mechanism in *Drosophila* flight muscles. The evidence for this hypothesis is five-fold. First, as soon as myofibrils have assembled, they are surrounded by mitochondria, which isolate each of them from their neighbouring myofibrils. Hence, direct mechanical contact between neighbouring myofibrils is blocked (Fig. 9). Second, when myofibrils and mitochondria mature, both strongly expand in diameter, generating an extensive mechanical communication interface between them. The ellipsoid mitochondria shapes along the myofibril axis together with the induced mitochondrial

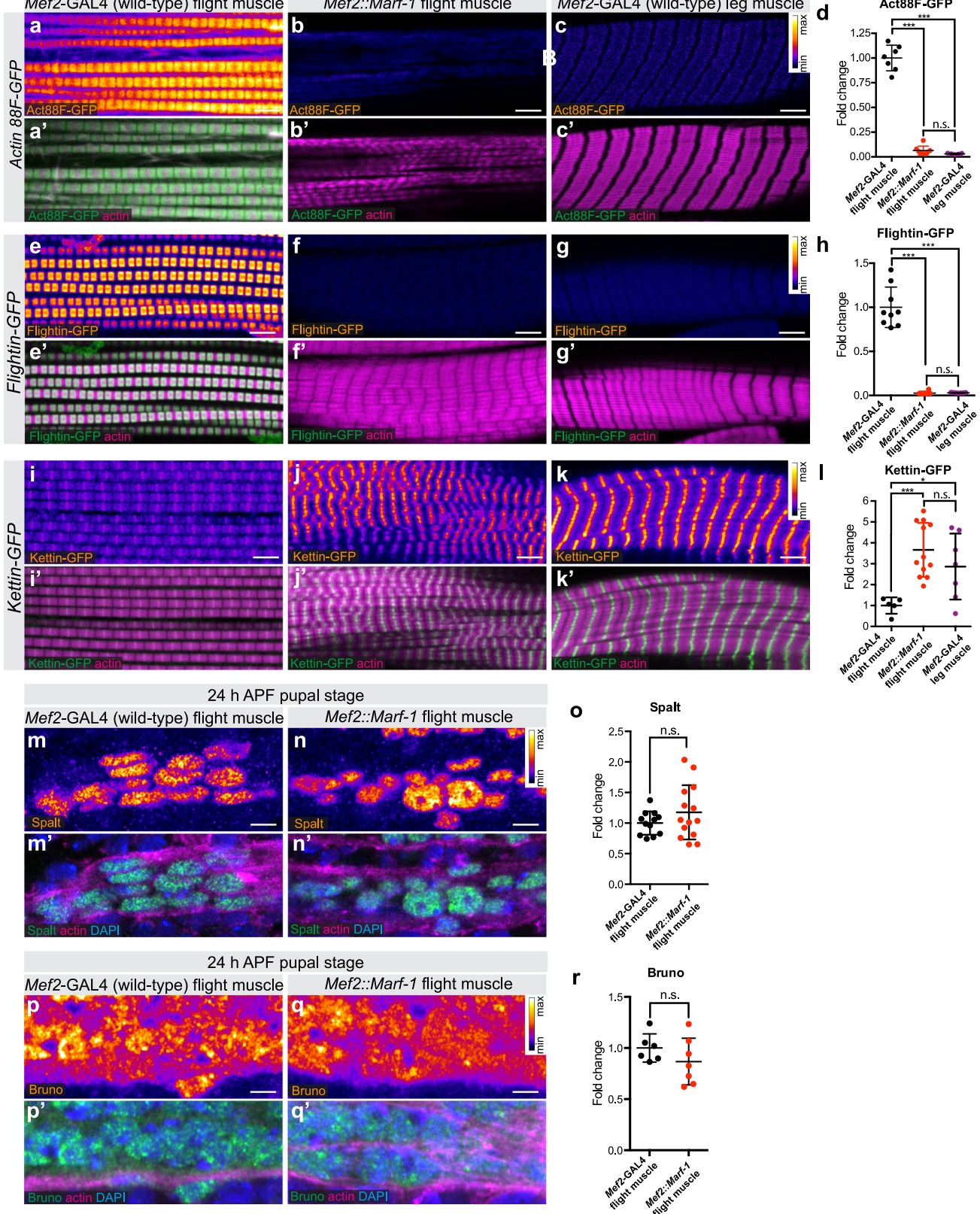

indentations caused by the myofibrils strongly support a role for mechanical pressure from myofibrils on mitochondria and vice versa. Third, in contrast to leg muscles, no specific contact sites at particular sarcomeric locations are present in flight muscles arguing against localised protein-protein-interactions mediating the spatial proximity and hence favouring the mechanical

interaction hypothesis. Fourth, relaxing mechanical constraints in the adult flight muscles by cutting myofibrils results in an immediate rounding of mitochondria, strongly suggesting that pressure directly shapes mitochondria. Finally, if intercalation is compromised, myofibrils grow larger in diameter, consistent with a mechanical feedback controlling myofibril diameter in flight

**Fig. 5 Mitochondria hyper-fusion causes a transcriptional shift to cross-striated muscle type. a–k** Adult wild-type (**a**, **e**, **i**) as well as *Mef2::Marf-1* flight muscles (**b**, **f**, **j**) and wild-type leg muscles (**c**, **g**, **k**) expressing GFP-tagged muscle-type specific proteins Actin 88F-GFP (**a–c**), Flightin-GFP (**e–g**) and Kettin-GFP (**i–k**); samples were fixed and actin was visualised with phalloidin. Relative GFP fluorescence levels are represented via a pixel intensity scale (white represents higher intensity). **d**, **h**, **l** GFP fluorescence was quantified with quantitative confocal microscopy (see Methods section) and plotted relative to control flight muscle levels (in **d** $n = 7$, 8 and 5 animals, respectively; in **h** $n = 9$, 6 and 7 animals, respectively; in **l** $n = 5$, 12 and 5 animals, respectively). Note that *Marf* over-expression in flight muscle converts the expression levels towards wild-type leg muscle levels. **m–o** Spalt protein levels in developing flight muscle myotubes at 24 h APF were quantified using immunostaining and quantitative confocal microscopy comparing wild type (**m**, **o**; $n = 12$ animals)) to *Mef2::Marf-1* (**n**, **o**; $n = 14$ animals). Actin was visualised with phalloidin, nuclei with DAPI. Note the comparable expression levels. **p–r** Bruno protein levels in developing flight muscle myotubes at 24 h APF were quantified using immunostaining and quantitative confocal microscopy comparing wild type (**p**, **r**; $n = 6$ animals) to *Mef2::Marf-1* (**q**, **r**; $n = 7$ animals). Actin was visualised with phalloidin, nuclei with DAPI. In all plots the mean ± standard-deviation (SD) is indicated, each dot the value from single animals, and significance from two-tailed unpaired *t*-tests is denoted as *p*-values = 0.0287 (*), ***$p \leq 0.001$. n.s. non-significant. Scale bars are 5 µm.

muscles. Together, these data strongly support a role for mechanical forces coordinating mitochondria and myofibril morphogenesis in flight muscle (Fig. 9).

Surprisingly, we have found that interfering with mitochondrial intercalation changes the transcriptional state of the flight muscles by down-regulating some flight muscle specific sarcomeric isoforms and up-regulating at least one cross-striated muscle-specific isoform (Fig. 9). Mechanistically, we showed that this change happens downstream of Spalt, since Spalt-dependent flight muscle specification is normal. How does the defective mitochondria intercalation feedback on transcription and also splicing as is the case for the Kettin-GFP? It has been shown that transcription is strongly regulated during myofibril maturation of *Drosophila* flight muscles resulting in a boost of sarcomeric gene expression[36,39]. Furthermore, it is well established that during mammalian muscle fibre-type maturation, sarcomeric isoform expression changes from embryonic isoforms, to neonatal ones and finally to adult isoforms[40,41]. How these switches in these different muscle types are controlled is not fully understood, but it is conceivable that changes in mitochondrial metabolism may contribute in both systems. Alternatively, as manipulating mitochondrial dynamics affects myofibril alignment in flight muscles, this change in the biomechanical properties of the myofibrils may impact the transcriptional status of the flight muscle fibre. Such a transcriptional feedback would ensure a direct coordination between mechanical and physiological requirements of the developing muscle fibres and thus may also be applicable to mammalian muscle fibres.

How do mitochondria intercalate between myofibrils? This is likely an active mechanism as it happens rapidly during a few hours of muscle development. A first explanation could be that the driving force can either originate by the assembly of myofibrils directly, which re-distribute throughout the fibre, starting from a more peripheral actin filament meshwork[19,36]. A second more attractive explanation would be an active mitochondrial transport mechanism, as mitochondria align along the axis of the newly formed myofibrils. Transport could be achieved by microtubule motors, since they have been described to transport mitochondria in various other cell types, particularly in neurons[42,43]. Interestingly, microtubules have been found in close proximity to the freshly assembled myofibrils in flight muscles[22,44] and hence are ideally placed to mediate mitochondria intercalation and alignment with the myofibrils.

Mechanical roles of mitochondria are not limited to muscle fibres. Pushing forces of polymerising actin filaments against a mitochondria network surrounding the spindle in mouse oocytes demonstrated a mechanical role of mitochondria in spindle positioning[45]. Also in this system, a fine balance between mitochondrial fusion and fission was necessary for normal spindle positioning[45]. Similarly, mitochondrial remodelling into long

giant mitochondria has been shown to be essential for sperm tail elongation during *Drosophila* spermatogenesis[46]. In these cells, mitochondria provide the platform for polymerising microtubules and the mechanical link between microtubules and mitochondria is essential for sperm tail elongation[46].

Muscle fibres contain very crowded cellular environments. Thus particularly in cardiomyocytes, which do contain large amounts of mitochondria[9] and share their high mechanical stiffness and high passive tension with flight muscles[47], the mechanical communication between mitochondria and myofibrils might be most prominent. However, we found here that even in cross-striated *Drosophila* leg muscles mitochondria do contact sarcomeres similarly to the contacts described for the 'intermyofibrillar' mitochondria in proximity to the sarcomeric I-bands that are more prominent in mammalian oxidative muscle[7]. Whether potential differences of mitochondria–myofibril interactions between oxidative and glycolytic mammalian fibres types impact their fibre-type-specific transcription will need to be investigated in the future. However, the documented differences in mitochondria morphology suggest that mechanical communication between myofibrils and mitochondria might be of general importance to successfully coordinate muscle development and homeostasis.

Interestingly, changing the fine-tuned fusion-fission balance of mitochondria results not only in severe muscle fibre phenotypes during mouse development[48,49], but also leads to severe impairment of muscle function and fibre loss if acutely manipulated in adult mice[50,51]. Furthermore, maintaining a healthy balance of mitochondrial fission and fusion is also essential to build and maintain a healthy mouse heart[52]. Interestingly, reducing mitochondrial fission results in dilated cardiomyopathy in neonatal mouse hearts, coinciding with impaired myofibril morphogenesis[53]. Even manipulating mitochondria dynamics after birth causes cardiomyopathies in mice[54]. Together, this highlights the importance of mitochondrial dynamics for muscle development and maintenance. While it is recognised that mitochondria networks are highly dynamic in healthy and diseased muscle fibres and cardiomyocytes, the here hypothesised mechanical coupling between mitochondria networks and the contractile machinery is still underappreciated in mammalian muscle.

## Methods

**Fly strains and genetics**. Flies stocks were maintained under normal culture conditions in humidified incubators with 12-h light–dark cycles. All fly stocks were maintained on standard lab fly medium. The standard lab medium is a variation of the Caltech media recipe, which includes 8% (w/v) cornmeal, 2% (w/v) yeast, 3% (w/v) sucrose, 1,1% (w/v) agar and 1% (v/v) acid mix. To prepare the media, cornmeal (80 g), sucrose (30 g), dry-yeast (20 g) and agar (11 g) were mixed in 1 L of water and brought to boil with constant stirring. The media was allowed to cool down to 60 °C, before 10 ml of acid mix was added. Acid mix was prepared by mixing equal volumes of 10% propionic acid (v/v) and 83.6% orthophosphoric

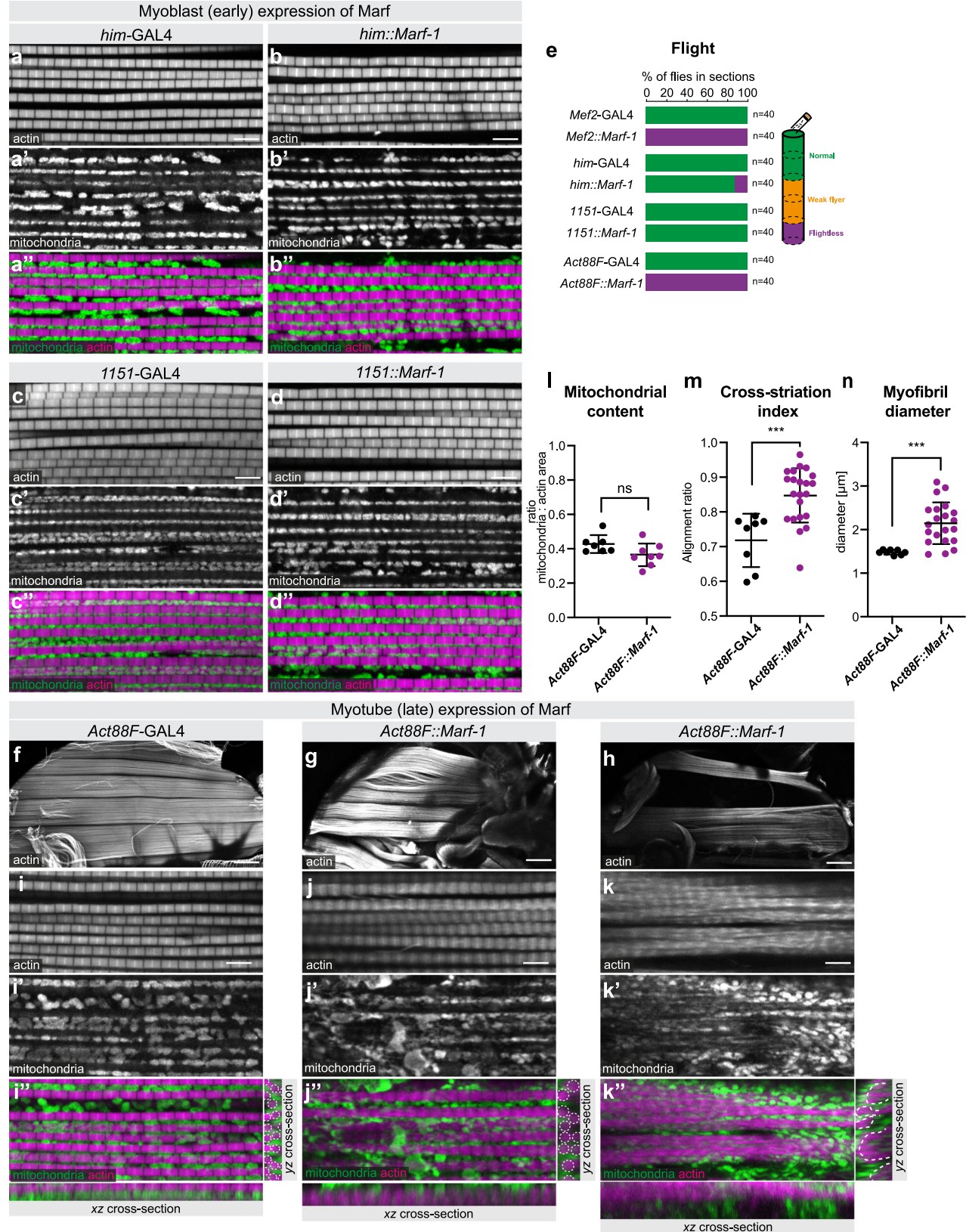

acid. The medium was then poured in vials (~10 ml/vial) or bottles (50 ml/bottle) and allowed to cool down before storing at 4 °C for later usage.

Wild-type control flies are either GAL4 driver crossed to *w[1118]* strain or the corresponding UAS lines crossed to *w[1118]*, as indicated in the figure legends. The strains used in this study are detailed in Supplementary Table 1. All crosses were developed at 21 °C in order to reduce GAL4 activity, unless otherwise mentioned.

Developmental times indicated are the equivalent to the characterised ones at 27 °C[36].

**Generation of UAS-MOM-GFP transgenic flies**. The *UAS-MOM-GFP* construct used in this study was generated by subcloning a gBlock (Integrated DNA Technologies) containing the *Drosophila* homologue of rat Tom20 minimal sequence

**Fig. 6 Developmental timing of mitochondrial dynamics impacts myofibril development. a–e** Wild-type adult control (**a**, **c**) and early over-expression of *Marf-1* with *him*-GAL4 (**b**) or *1151*-GAL4 (**d**) flight muscles were stained with phalloidin and anti-complex V antibody to visualise myofibrils and mitochondria. Note the normal myofibril and mitochondria morphologies (**b**, **d**), which support flight (**e**). **f–n** Wild-type control (**f**) and *Act88F::Marf-1* hemi-thoraces (**g**, **h**), as well as flight muscles (**i–k**) expressing *Marf-1* during later developmental stages were stained with phalloidin and anti-complex V antibody to visualise myofibrils and mitochondria. Two representative phenotypes of *Act88F::Marf-1* flight muscles are shown, displaying either thicker myofibrils (indicated by white dashed outline in the cross-section) (**j**) or partially cross-striated myofibrils (**k**). Mitochondria are largely excluded from myofibrillar bundles in *Act88F::Marf* muscles (see dashed white outline of the myofibril-rich areas in **k″**). **l–n** Quantification of the *Act88F::Marf* flight muscle phenotypes, quantifying mitochondrial content (**l**; $n = 7$ and 8 animals, respectively), cross-striation index (**m**; $n = 8$ and 21 animals, respectively) and myofibril diameter (**n**; $n = 8$ and 21 animals, respectively). In all plots the mean ± standard-deviation (SD) is indicated and significance from two-tailed unpaired *t*-tests is denoted as ***$p$-values $\leq 0.001$. n.s. non-significant. Scale bars are 5 μm (**a–d**, **i–k**) and 100 μm (**f–h**).

determined as being sufficient for mitochondria outer membrane (MOM) targeting[55].

(ATGATTGAAATGAACAAAACTGCAATCGGCATTGCAGCGGGAGTA GCTGGAACTCTGTTTATTGGATACTGCATCTACTTCGACAAGAAGCGC CGCAGCGATCCCGAGTACAAGAAGAAAGTCCGT), fused in frame to sfGFP into pUASt-attB vector[56] with EcoRI and NotI. The resulting plasmid was sequenced with the following primers: 5′-[GCAGGCCGAATTCATGATTG]-3′ and 5′-[CGTGGTCAGCCATTAGAATG]-3′ and integrated into attP site VK00033 using standard methods[33] (see Supplementary Table 1).

**Flight and leg muscle preparations, immunohistochemistry**. Preparations of adult hemithorax or fixed pupa for microscopy have been described in detail[34]. Briefly, pupae were developed and staged at 21 °C, and fixed by paraformaldehyde 4% in PBS with 0.3% Triton X-100 (PBS-T), for 25 min at 31 h, 42 h or 62.5 h after pupa formation (APF), which correspond to 24 h, 32 h or 48 h APF of development at 27 °C, respectively[36]. Actin was labelled with phalloidin-rhodamine (1:500; Molecular Probes) and nuclei with DAPI. Mitochondria were labelled by the expression of GFP fused to a mitochondrial matrix (mito-GFP) or outer membrane signal (MOM-GFP) and detected by direct fluorescence without staining. Alternatively, mouse anti-complex-V (anti-ATP5a, Abcam ab14748; 1:500) immunostaining was used to label mitochondria. Spalt protein was visualised with anti-Spalt antibodies (gift of Reinhard Schuh, 1:100) and Bruno protein was visualised with rat anti-Bruno antibodies (gift of Anne Ephrussi, 1:1000). Following 3 × 10 min washes with PBS-T, samples were incubated with secondary antibodies carrying the indicated fluorophores (Alexa Fluor 488 goat anti-mouse and Alexa Fluo 488 goat anti-rabbit, both at 1:1000) for 2 h at room temperature (20–22 °C). GFP-fosmid lines, as indicated in the fly strains table, were used for direct visualisation of Kettin-GFP leg isoform, Flightin-GFP and Act88F-GFP protein levels[33]. Samples were washed twice with PBS-T (5 min) and mounted in VectaShield containing DAPI using two cover slips as spacers. All reagents used are detailed in Supplementary Table 1.

**Flight**. Flight tests were performed as previously described[57] and as reproduced here. Twenty males 1-week-old were collected at least 24 h prior to testing and then dropped into a 1-m-long transparent plexiglass tube with 8 cm inner diameter, divided into five zones. Those that landed in the top two zones were considered 'normal fliers', those in the next two zones 'weak fliers' and those that fell to the bottom of the cylinder 'flightless'. In crosses with GAL4 insertions on the X chromosome females were used. Tests were repeated at least twice per genotype, for a minimum of 40 flies in total per condition.

**Live dissection to visualise flight muscles without fixation**. Living hemithorax were dissected and mounted in Schneider medium. Living samples were imaged within 30 min following dissection. Dissection consisted of a precise incision through the cuticle with sharp forceps (#11252-20 Dumont#5, Fine Science Tools) at the median plane resulting in the separation of the two hemi-thoraces. Ventral connective tissues were cut along the midline into two halves using fine dissection scissors (#15009-08 Fine Science Tools) to completely detach left from right halves. The dissection is usually non-invasive for the flight muscle resulting in intact flight muscle fibres attached to the tendon cells of the thorax. Samples were mounted in Schneider medium using two cover slips as spacers and imaged immediately.

**Cross-striation index**. To quantify the vertical alignment of individual myofibrils we a defined a 'cross-striation index' as the ratio between the distance needed to connect M-bands from neighbouring myofibrils and a straight line perpendicular to the myofibril horizontal axis from first to last myofibril used for quantification. To avoid bias, the nearest M-band was chosen when a horizontal path needs to be made in between myofibrils. Perfect alignment results in a ratio of 1 and lower values represent progressively less alignment (see Supplementary Fig. 1).

**Sarcomere quantification**. Sarcomere length and diameter quantification were made using the MyofibrilJ plugin for Fiji (https://imagej.net/MyofibrilJ)[36,58]. For

genetic interventions that result in strong sarcomere phenotypes, which cannot be analysed automatically by the plugin, measurements were made manually. An average of ten myofibril diameters per sample was then made on an interpolated *YZ* projection using Fiji. If samples showed a strong cross-striation phenotype, they were not quantified for myofibril diameter, but instead included for the cross-striation index quantification.

**Mitochondria content quantification**. Total areas of actin (phalloidin) or mitochondria (mito-GFP) were identified via Otsu thresholding in Fiji for each individual acquisition channel. This was done for each *Z*-plane and channel, and thresholding was reset for each new plane in the same image, to correct for signal loss due to section depth. Multiple quantifications (the entire *Z*-stack at multiple *XY* ROIs per fly) from single flies were averaged and plotted as a single value in the figure plots (each fly counts as $n = 1$).

For tubular leg muscles and *Marf* gain of function flight muscles, an interpolation across the *Y*-axis was made to generate a new image in which the *Z*-axis becomes the longitudinal axis of the tube (*Z*-depth cross-section). This new stack was then segmented for each slice (1024 in total for each image stack) to distinguish the peripheral area of the tube – rich in actin – from the central area where a higher amount of mitochondria and no actin signal are present. Signal from either channel, actin or mitochondria, was then quantified as described above.

**Individual mitochondrion area**. Mitochondria signal from mito-GFP expressing adult flight muscles was used to individually segment each mitochondrion across all *Z*-planes from the entire *Z*-stack. Fiji was used to apply a Gaussian filter (lambda = 2), background correction and Otsu thresholding, followed by watershed on the binary image. Individual objects were quantified for area (6894 for wild type and 14,443 for *Mef2::Marf-IR*).

**3D reconstruction and analysis**. High-resolution confocal imaging was performed using a Zeiss LSM880 confocal microscope equipped with an Airyscan detector. Mitochondria were visualised in flight and leg muscle with two different labels: (1) mitochondria matrix mito-GFP or (2) mitochondrial outer membrane MOM-GFP. Flight muscle mitochondria were then segmented with a machine learning algorithm described in detail below, leg muscle mitochondria were segmented with the Fiji plugin "Interactive Watershed" (https://imagej.net/Interactive_Watershed). This plugin allows for extensive manual optimisation of objecting splitting/joining in large stacks and includes 3D water-shedding, essential for our 3D reconstruction. Continuous validation for the watershed splitting was verified manually and we opted to have more splitting than to have too many large objects (by missing splitting). This compromise led to some network connections between mitochondria being missed.

The resulting binary images, for both tissues, were then connected in 3D via the MorpholibJ plugin for Fiji using 'connected components labelling'[59]. Size Open (min 100 voxel filter) was applied and objects on the borders of the 3D space were discarded, and a 26 voxel connectivity used between *Z*-slices. This was followed by 3D object analysis in MorpholibJ, from which we obtained individual volumes used to colour code the mitochondria in the volume renderings, as well as the ellipsoids, long, medium and short axis and azimuts. 3D visualisation was done with the 3D Viewer plugin from Fiji and with Amira Software (Fisher Scientific).

**Deep Learning segmentation**. A Deep Learning model has been trained to segment mitochondria labelled with MOM-GFP in the flight muscle. A Github Repository containing the code used for the Deep Learning Segmentation and the Shape classification (section below) and quantification analysis, including a detailed Readme file and notes directly into the notebooks, can be found at https://github.com/fabda/Myofibril_paper.

To segment flight muscle mitochondria from confocal microscopy data a fully convolutional encoder-decoder UNET architecture has been used with an ImageNet pretrained seResNet18 encoder as backbone[60,61]. To train the model, we built a relatively small dataset consisting of only 13 pairs of 128 × 128 image tiles extracted from one part of the entire 1024 × 1024 image stack and their corresponding manually drawn masks with Fiji. To perform strong network

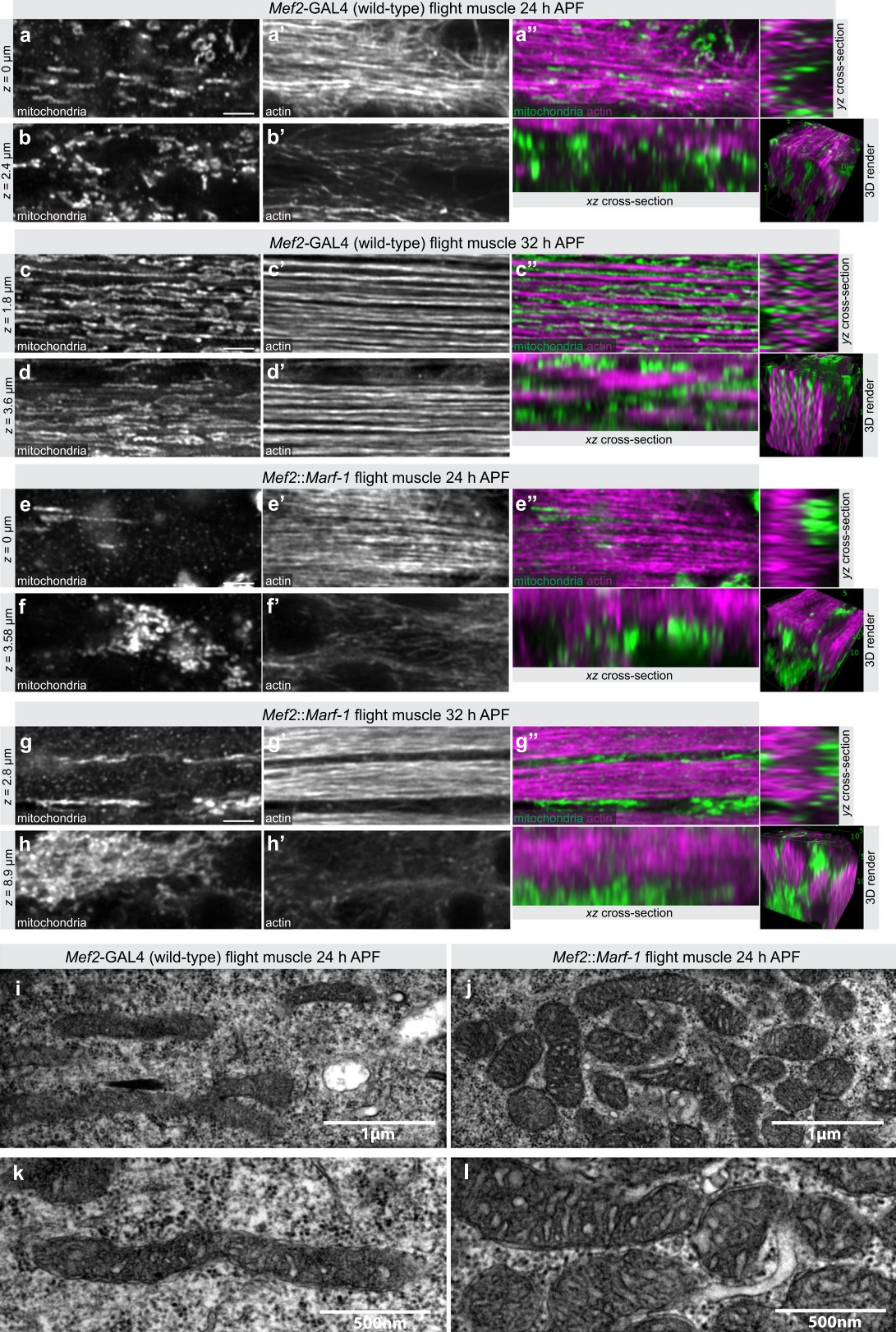

**Fig. 7 Developmental effect of mitochondria hyper-fusion. a–h** Developing wild-type flight muscles at 24 h after puparium formation (APF) (**a**, **b**) and 32 h APF (**c**, **d**), compared to *Mef2::Marf-1* flight muscles at 24 h APF (**e**, **f**) and 32 h APF (**g**, **h**). See also Supplementary Movie 8. Mitochondria were visualised with mito-GFP and actin with phalloidin. Note the mitochondrial intercalation between myofibrils in wild-type 32 h APF flight muscles (**c″**), which is blocked by *Mef2::Marf-1* (**g″**). **i–l** Developing wild-type (**i**, **k**) compared to *Mef2::Marf-1* (**j**, **l**) flight muscles at 24 h APF visualised by electron microscopy (EM). Two representative regions at different magnifications are shown. Note the clustered and more rounded mitochondria upon *Mef2::Marf-1*. Scale bars are 2.5 μm (**a–g**) or directly noted on EM images (**i–l**).

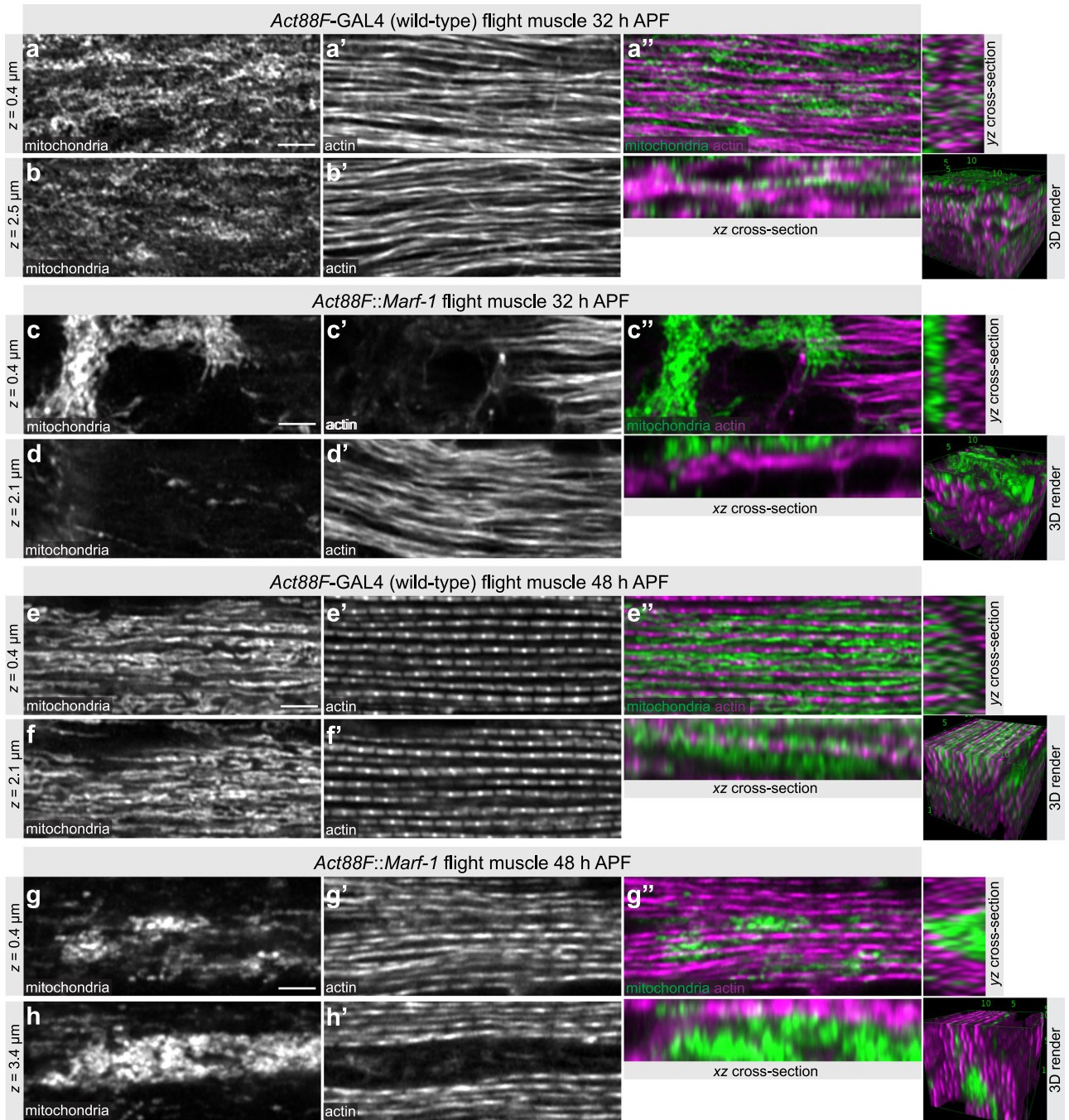

**Fig. 8 Mitochondria isolate individual myofibrils. a–h** Developing wild-type flight muscles at 32 h after puparium formation (APF) (**a**, **b**) and 48 h APF (**e**, **f**), compared to *Act88F::Marf-1* flight muscles at 32 h APF (**c**, **d**) and 48 h APF (**g**, **h**). See also Supplementary Movie 9. Mitochondria were visualised by immunostaining against complex V (ATPase) and actin with phalloidin. Note how mitochondria isolate myofibrils in wild type (**e**″) but fail to do so in *Act88F::Marf-1* with mitochondria clustering centrally (**g**″). Scale bars are 2.5 μm.

regularisation in order to increase model performances despite the small dataset size, a data augmentation approach was successfully applied to virtually increase the training set size by generating, for every epoch (20 epochs in total), 400 batches of 32 artificially generated 128 × 128 image tiles by using a combination of random horizontal/vertical flips, width/heights shift and zooms of the original training dataset. Dice-Sörensen loss function were chosen to optimise the network weights with Adam optimizer (learning rate 1e-4) and Intersection Over Union metrics has been used to assess segmentation quality (0.85 IoU on validation set). The model has been trained using Python Keras framework (version 2.2.4) with Tensorflow (version 1.15) as backend on one Nvidia Quadro GV100 GPU card. Each slice was segmented individually by splitting every slice into 64 adjacent tiles of size 128 × 128 to feed the trained model, retrieve their predicted segmentation mask and recombine everything to achieve the whole slice and volume segmentation.

The resulting binary images were then connected in 3D via the MorpholibJ plugin for Fiji as described above (see Supplementary Fig. 2a, b).

To segment the serial block-face electron microscopy acquisitions in the leg muscle, a large crop from the original acquisition was used for carefully curated hand segmentations, to establish the ground truth and generate the classification model. Briefly, a stack of 100 slices, 30 nm apart, and of 704 × 704 pixels each (7040 × 7040 nm, for a total volume of 149 μm³) was used to generate a set of ROIs with ImageJ ROI Manager that filled the mitochondria area, for all individual slices in the stack. A total of 2021 objects were segmented covering 7,979,577 pixels.

This 704px × 704px × 100px hand segmented stack plus the original corresponding raw stack was then used for the training/validation of a Deep Learning segmentation model using Keras python framework 2.2.4 with Tensorflow Backend 1.15 and segmentation models library[62]. First, a data

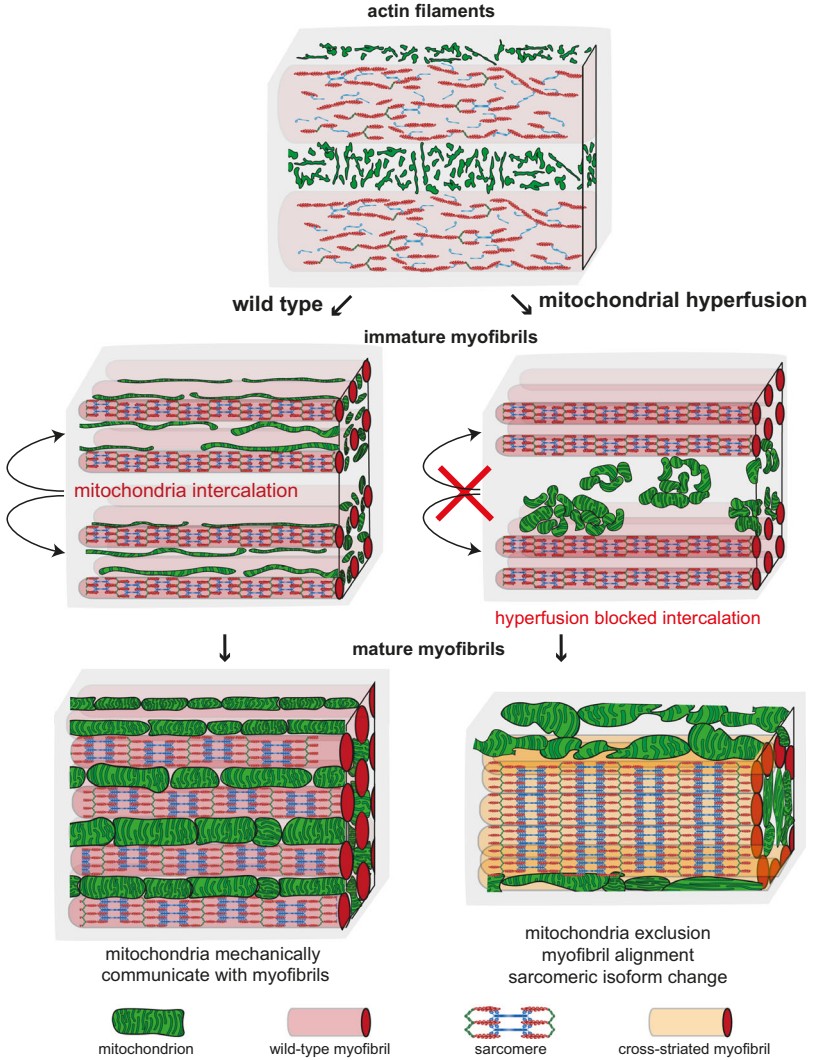

**Fig. 9 Mitochondria–myofibril communication model.** Developing flight muscle schemes to highlight the interplay between mitochondria with actin filaments (top, early stage), immature myofibrils (middle, intermediate stage) and mature myofibrils (bottom, mature stage). Wild type is shown on the left and mitochondrial hyper-fusion on the right. At the actin filament stage, mitochondria display a filamentous network morphology spatially separated from the actin filament mesh. Upon myofibril assembly, mitochondria intercalate between myofibrils in wild type and establish a tight mechanical communication. Myofibril and mitochondrial diameter growth causes generation of mechanical pressure and isolates individual myofibrils. Mechanical feedback ensures the correct myofibril diameter. In contrast, hyper-fusion of mitochondria results in larger clustered mitochondrial networks that fail to intercalate between the immature myofibrils. As a consequence of mitochondrial exclusion, the mechanical communication between mitochondria and myofibrils is limited and myofibrils align with each other around centrally clustered mitochondria. Scheme for sarcomeric components was adapted from Lemke and Schnorrer[3].

augmentation strategy known as "random cropping" was applied on it in order to increase the dataset size by randomly cropping smaller 256 × 256 tiles out of the 704 × 704 × 100 stack and then building a 50k training/validation image/mask dataset. The segmentation model developed is based on a U-Net architecture using a pretrained ResNet152 encoder as backbone, Adam optimizer (learning rate tuned progressively by hand from 1e-3 to 1e-5 during training), a combined binary cross entropy plus Jaccard coefficient as network loss function, sigmoid as final activation layer and the IoU (Intersection Over Union) metrics to monitor the model performance (0.98 IoU on validation set). Once trained, the model has been used to segment the full size stack. To do so, first, the full size stack has been tiled into small non overlapping adjacent 256 × 256 tiles to fit with our segmentation model input shape and then for each of them we applied the segmentation model to generate their 256 × 256 corresponding predicted segmentation mask. Finally, we recombined all predicted masks to create the corresponding full size segmentation mask.

**3D object classification.** Classification of objects as spheres, ellipsoids or rods was performed according to Isaac[63]. Elongation descriptor was calculated as the length of object's ellipsoid length divided by half of the sum of its width and thickness. Flatness descriptor was obtained by dividing object's width by its thickness. Sphere

class was associated to every object having both their elongation and flatness <1.3; rod class was assigned to object having an elongation superior to 2.5; and all remaining objects were assigned to the ellipsoid category. The classes were used to colour code individual mitochondria and we verified for their accuracy by going through the stacks manually.

**Statistics and reproducibility.** For all experiments at least two independent experiments were conducted, which showed reproducibility of the phenotype as reported. Each sample quantified relates to an individual animal, thus every sample is a biological replicate, and panels show an aggregate of all independent experiments quantified. Detailed information on the number of animals/samples used for each quantification shown in the figures, as well as the statistical tests and p-values, is presented in the supplementary Source Data file.

**Serial block-face scanning electron microscopy.** One-week-old *Drosophila* thoraxes were dissected rapidly in cold PBS and immediately fixed in 2% paraformaldehyde, 2.5% glutaraldehyde overnight at 4 °C. Samples were contrasted for 1 h in potassium ferrocyanide-reduced osmium (2%), 20 min in TCH, then 30 min in 2% osmium before being incubated at 4 °C overnight in 1% uranyl acetate.

Between each incubation, five times 3 min washes in water were done. The samples were incubated in fresh lead aspartate at 60 °C for 30 min and then dehydrated by a series of 10 min incubations in 20%, 50%, 70%, 90%, 100% ethanol and finally by glass-distilled acetone twice. Then, the samples were incubated in increasing concentrations of resin for 2 h each at room temperature (Durcupan 25%, 50%, 75 and 100% diluted in acetone). The Durcupan 100% is renewed and incubated for 16 h, then incubated for 48 h at 60 °C for the polymerisation of the resin. Details are described in the NCMIR protocol for SBF-SEM[56]. Imaging was carried out on a FEI Teneo VS running in low vacuum (30 Pa), at 2 kV and using a backscattered electrons detector. The acquisition voxel size was $5 \times 5 \times 40$ nm (flight muscle dataset) or $10 \times 10 \times 30$ nm (leg muscle dataset). The segmentation was carried out manually in IMOD or automatically as described above.

**Transmission electron microscopy**. Pupae were collected after 24 h APF equivalent. The pupal case was removed and three holes punctured in the abdomen to facilitate diffusion of fixation reagents. Fixation was done with 2% paraformaldehyde, 2.5% glutaraldehyde in 0.1 M sodium cacodylate overnight at 4 °C. Abdomen and head were cut manually with a microtome blade and the single thoraces were contrasted with osmium tetroxyde (2%) in 0.2 M sodium cacodylate for 2 h 30 min on ice. Samples were washed for 10 min with 0.1 M sodium cadodylate on ice, followed by an overnight incubation in 2% uranyl acetate in $H_2O$ at 4 °C. Samples were dehydrated and imbedded in resin similarly as described above for the SBF-EM protocol, with all steps done on ice. Thoraces imbedded in resin were then cut in ultra-thin sections of 75 nm using a microtome from the dorsal towards the ventral side. Samples cut in ultra-thin slices were contrasted a second time with 2% uranyl acetate for 5 min followed by incubation for 2.5 min in a freshly made lead citrate solution (to 16 ml of $ddH_2O$ add 3 ml 1 M lead nitrate, mix, add 2 ml 1 M sodium citrate, mix gently and finally add 4 ml 1 M sodium hydroxide drop by drop). Imaging was carried out on transmission electron microscope FEI Morgagni 120 kV.

**Nuclei quantification**. In all, 24 h APF *Drosophila* pupae were fixed with 4% paraformaldehyde during 20 min and then bissected from the dorsal towards the ventral side along the length to render two mirror halves, exposing the developing IFM in the thoraces. The fixed hemi-pupae were stained with rhodamine-phalloidin to mark the actin filaments and with DAPI to stain the nuclei. Fused nuclei were quantified by manually counting DAPI stained nuclei in a fixed volume of 11,000 $\mu m^3$ for every acquisition for both control and *Mef2::Marf1* conditions. A 3D reconstruction stack (with an equal number of slices for every sample) allowed to exclude nuclei already counted on previous slices, as well as nuclei located on borders. The phalloidin stain facilitated identifying the nascent myotubes and exclude non-fused nuclei.

**Reporting summary**. Further information on research design is available in the Nature Research Reporting Summary linked to this article.

## Data availability
The gene expression data of indirect flight muscle development used for Supplementary Fig. 6 was published previously[36] and is available at NCBI's Gene Expression Omnibus (GEO) with the accession number GSE107247.

The authors declare that the data supporting the findings of this study are available within the paper and its supplementary information files as a supplementary. Any generated Drosophila stocks and any other data that support this publication are available from the corresponding author upon reasonable request. Source data are provided with this paper.

## Code availability
A Github Repository containing the code used for the Deep Learning Segmentation and the Shape classification (section below) and quantification analysis, including a detailed Readme file and notes directly into the notebooks, can be found at https://github.com/fabda/Myofibril_paper.

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

## Acknowledgements

The authors are indebted to the IBDM imaging facility for help with image acquisition, maintenance of the microscopes, 3D visualisation and generating animations and the Turing Centre for Living Systems engineering team for their help with data analysis. The electron microscopy experiments were performed on the PiCSL-FBI core facility with help from Aicha Aouane. We acknowledge the France-BioImaging infrastructure supported by the French National Research Agency (ANR-10–INBS-04-01, Investments for the future). The authors are grateful to Christophe Pitaval and Céline Guichard for fly embryo injections, to Julien Royet for supporting parts of this work in his lab and to Flybase for database organisation. Fly stocks obtained from the Bloomington Drosophila Stock Center (NIH P40OD018537) and the Vienna Drosophila Resource Center (VDRC) were used in this study. The authors thank Bianca Habermann, Pierre Mangeol, Qiyan Mao and Clara Sidor for helpful discussions and critical comments for this manuscript. This work was supported by the European Research Council under the European Union's Seventh Framework Programme (FP/2007–2013)/ERC Grant 310939 (F.S.), the Centre National de la Recherche Scientifique (CNRS, F.S., N.M.L.), the excellence initiative Aix-Marseille University A*MIDEX (ANR-11-IDEX-0001–02, F.S., C.R.), Aix-Marseille University (PhD fellowship. J.A.), the French National Research Agency with ANR-ACHN MUSCLE-FORCES (F.S.) and MITO-DYNAMICS (ANR-18-CE45-0016-01, F.S.), the Human Frontiers Science Program (HFSP, RGP0052/2018, F.S.), the Bettencourt Foundation (F.S.), the France-BioImaging national research infrastructure (ANR-10-INBS-04-01) and the Turing Center for Living Systems (CENTURI, A*MIDEX, Investments for the Future). The funders had no role in study design, data collection and analysis, decision to publish, or preparation of the manuscript.

## Author contributions

J.A., N.M.L. and F.S. conceptualised and designed the study; J.A. performed most of the experimental work and image acquisition with help by C.R. for the developmental data and N.B. for the electron microscopy; J.A., N.B., N.M.L. and T.R. generated the serial-block-face electronic microscopy data; F.D. and N.M.L. performed the 3D reconstruction analysis; F.D. developed and implemented the machine learning segmentation analysis; J.A., N.M.L. and F.S. analysed the data and prepared the figures; N.M.L. and F.S. wrote the manuscript with input from all authors.

## Competing interests

The authors declare no competing interests.
