## [Peer Review File · Nature Communications]

REVIEWER COMMENTS

Reviewer #1 (Remarks to the Author):

This manuscript focuses on the relationship between myofibril architecture and mitochondrial structure in the *Drosophila* muscle system. The data are quite interesting in that changing mitochondrial morphology in the IFM yields structural changes in the myofibrils that apparently arise from the production of non-IFM isoforms of contractile proteins. The changes cause the fibrillar nature of the muscle to be transformed toward a tubular muscle type, such as leg muscle, with adjacent myofibrillar elements yielding a more dramatic cross-striation pattern along with centrally located mitochondria, rather than the intercalated pattern typical of IFM. The authors have developed new methodologies for assessing mitochondrial morphology and present excellent images to back up their conclusions. The movies are particularly striking. Statistical work appears complete and data are presented in detail in a supplementary table.

Specific results show that genetically reducing mitochondria size prevents normal IFM function, but does not affect the level of cross striation. In contrast, enhancing mitochondrial fusion changes mitochondrial location and level of cross striation toward the tubular muscle phenotype. This change occurs at the transcriptional level for at least three contractile protein isoforms and mitochondrial fusion needs to occur during myofibrillogenesis for the effect to be seen. Developmental studies are nicely done to show how mitochondrial intercalation occurs and how it is compromised in *Marf1* IFM. Overall, the manuscript highlights a new area of investigation, the effect of mitochondrial morphology on myofibril organization, which will be of strong interest to individuals studying muscle structure, function and disease.

There are a number of areas that need to be improved:

1. When mentioning mechanical tension of myofibrils as causative of mitochondrial morphology, does this correspond to tension induced by actin-myosin interaction? Does removing this interaction genetically or with drugs affect mitochondrial morphology? Does removing thick or thin filaments affect mitochondrial morphology? The latter question likely can be answered by reference to the literature.
2. The introduction to *Spalt* knockdown in IFM on pages 6-7 does not adequately reference that this work was previously published, i.e., it reads as if it is a new finding. Perhaps the mitochondrial morphology was more closely examined here? In any case, reference 17 should be more specifically cited for the original discovery.
3. What about genes downstream of *Spalt*, since *Spalt* levels are normal? For instance, is *Arret/Bruno* splicing factor regulated? This seems an important question to address regarding mechanism.
4. On page 10: "over-expression of *Marf* with *Mef2-GAL4* results in fewer flight muscles, likely due to muscle atrophy during development" Do the authors have any data on this issue to bolster their

contention? Other possibilities exist, such as failure to form fibers or fusion of existing fibers.

5. Do the authors know what actin isoform is expressed when Marf is over expressed in IFM? This would be interesting to find out. Specific muscle form? Cytoplasmic form?

6. Does kettin in the IFM arise from alternative splicing or from using a different transcriptional start site, i.e., is the change due to splicing or transcription? This is related to the question as to whether only transcriptional changes (actin, flightin) occur or whether alternative splicing is altered as well.

7. Using the Actin88F promoter to express Gal4 and marf-1 is of some concern, since Actin88F production is turned down dramatically when marf-1 is over-expressed. Thus, one would think that Gal4 expression would be reduced as well. Does this explain the reduced phenotype compared to Mef2-Gal4?

8. Were DVMs and DLMs studied? Do they show any developmental differences in mitochondrial localization? Do the mitochondrial fusion and fission genes differentially affect them?

9. Can the authors be more specific about the leg muscles examined within the Results section? Which leg? Which type of leg muscle?

Minor Concerns:

1. On page 8: "To segment the complex shapes of leg muscle mitochondria we used the mitochondrial matrix mito-GFP marker expressed with Mef2-GAL4 (Fig. 2j,k, Supplementary Movie 3)." However, panel 2j refers to IFM.

2. On page 9: "Strikingly, severing myofibrils results in a dramatic rounding up of all neighbouring mitochondria into spheres (Fig. 3c,e). This transition was observed with both mitochondrial markers, MOMGFP (Fig. 3c) as well as with mito-GFP (Fig. 3e)." References to Fig. 3e should actually reference Fig. 3e'.

3. On page 10: "which adopt a spherical instead of an ellipsoid shape (Fig. 4d,e,k)" should reference 4d' and 4e'

4. Fig. legend 1: "Dotted lines in cross sections in (c'') and (h'') represent the regions measured." Should this be c''' ?

5. Fig. 2 panel h is not labeled.

6. Fig. 2 legend: "intimated contact": I presume the authors mean "intimate contact" Also, panels n, o, o' and o'' are not correctly cited in the legend.

7. Fig. 5 legend: "GFP fluorescence levels are represented via a heat map (fire LUT, white is higher)". I am not seeing a heat map and I don't know what the notation in parentheses denotes.

8. Fig. 6 legend (and paper text) regarding Act88F driver: “largely cross-striated myofibrils (k)”. This description appears to be an over-statement.

9. Figures 7 and 8 appear somewhat redundant. Perhaps they can be combined?

10. In both the introduction and discussion, more effort should be made to differentiate vertebrate and fly data. The reader may not appreciate that there are differences between the systems. For instance: “more elongated mitochondria present in slow fiber types” is presumably referring to vertebrate muscle only; “How these switches are controlled is not fully understood, but it is conceivable that changes in mitochondrial metabolism may contribute [talking about mammals]. Alternatively, as manipulating mitochondrial dynamics affects myofibril alignment, this change in the biomechanical properties of the myofibrils may impact the transcriptional status of the muscle fiber [talking about flies and possibly applying this to mammals].”

11. The use of “flight muscles” instead of “indirect flight muscles” or IFMs could prove confusing to the reader, as direct flight muscles are not studied here. Being more precise throughout or at least indicating the shorthand labeling of IFMs as flight muscle at the beginning of the manuscript would be important.

12. The term “cross-striated muscles” can be confusing, since all insect muscle is considered striated. While the leg muscle pattern is dramatic as opposed to the poorly aligned fibrillar nature of IFM, the authors use a striation index that shows a positive value in IFM. Hence this issue needs to be more carefully explained.

Reviewer #2 (Remarks to the Author):

The present paper addressed an interesting aspect of muscle morphogenesis related to the cross-talk between mitochondria and myofibrils. Initially, the authors characterized the morphology of mitochondria and myofibrils in leg and flight muscles and showed that the leg muscles displayed a cross-striated myofibrils while flight muscle a fibrillar morphology. This myofibrillar morphology with densely packed ellipsoid-like mitochondria at the periphery was dependent by the transcription factor Spalt. Beautiful 3D reconstruction images of mitochondrial networks underline the differences of mitochondrial shape between flight and leg muscles. Due to these observations and to the finding that a drop in mechanical tension results in mitochondrial shape change from ellipsoid to spheres losing the contact site with myofibrils, they postulated that mitochondrial dynamics may contribute to myofibrils morphogenesis. Gain and loss of function experiments for the fusion and fission mitochondrial proteins confirm that the shape of the network is important for the flight muscle function and for the mitochondrial intercalation between myofibrils. The work is well described and the images of mitochondrial network of leg and flight muscles are beautiful. The take home message is of interest for the community and further underline the critical role of mitochondria as hub of signaling pathways that retrogradely modulate genes transcription. However, some aspects are merely descriptive and need more insights. The following points should be considered.

Point 1. The idea that mitochondrial network may drive the morphogenesis of myofibrils is

interesting. However, the data that Marf knockdown of DRP1 overexpression does not change the fibrillar morphology (despite mitochondria are now rounded that is against a mechanical tension effect on mitochondrial shape) while Marf overexpression does, needs a rationale for the correct interpretation of this experiments that is to monitor Marf and Drp1 expression in leg and flight muscle before and after 24 h APF.

Point2. Images of mitochondrial-myofibrils network of MARF overexpressing flight muscles does not show the same quality of the initial figures. 3D reconstructions (including EM) should be also produced for these experiments (Fig 6-7-8).

Point3. Mitochondrial network was not shown when dnDRP1 was expressed while it should (Suppl Fig 3h-l). Blocking fission does not mean that the network is identical to the one of Marf overexpression.

Point4. The atrophy observed in Marf1 and dnDRP1 overexpressing flies is problematic because per se it can interfere with myofibril morphogenesis. Is this atrophy due to increased protein breakdown?

Point5. Act88F, Flightin and Kettin expression should be monitored also in Act-88F-Marf-1 line.

Point6. It would be important to understand which step downstream Spalt is regulated by Mitochondrial dynamics

Reviewer #3 (Remarks to the Author):

This is an outstanding paper investigating the synergistic development of muscle mitochondria and myofibrils. The paper combines sophisticated imagery with cutting edge computer aided quantitative analysis, the power of Drosophila genetics and mutants, and leverages the contrasting properties of two Drosophila muscle types, fibrillar versus cross-striated. The images and videos are not only very informative but entrancing and beautiful.

Some of the new and novel findings that significantly advance our understanding of muscle development include: In fibrillar muscle where each myofibril is surrounded by mitochondria, mechanical pressure from the myofibrils force mitochondria into an elongated shape that causes a high degree of contact between the two. This intimate contact presumably helps with delivery of ATP to myofibrils. Conversely, the elongated mitochondrial that are squeezed between the myofibrils, help prevent individual myofibrils from fusing together to form cross-striated myofibrils. One method by which this was shown was by genetically increasing fusion rate to prevent long intercalated myofibrils. This also helps regulate myofibril size. The authors show that cross-straited muscle forms, at least in part, due to higher fusion rate of mitochondria.

The paper includes excellent use of Drosophila genetic to show that altering mitochondria formation, particularly fusion, leads to converting non-striated muscle to a striated muscle morphology, i.e. by disrupting mitochondrial intercalation. This conversion included altering expression of isoform specific proteins of the myofibril, which raises the very interesting question of how this feedback occurs.

The methods all appear to be sound and I did not see any problems. I particularly liked the

mechanical disruption of tension on myofibrils and mutants that increase or decrease mitochondrial fusion.

Good controls are included such as altering Spalt expression levels, and appropriate use of flight tests to verify that the results are physiologically important. Interpretation also appears quite sound and backed by strong experimental results.

I think this paper will be of high interest to researchers in several fields including developmental biology, mitochondria biogenesis, muscle energetics and muscle mechanics. I predict it will be very well received.

Minor concerns:

- 1) In the discussion, the authors could consider how their findings with *Drosophila* relate to the different muscle types found in mammals that they mentioned in the introduction. The intro mentions higher endurance muscles (which presumably includes slow-twitch fiber types?) have more elongated mitochondria. Do these types also have less fused mitochondria than fast twitch? The authors include a good discussion of heart muscle, but how about skeletal? What type of myofibrils do fast and slow twitch have? Cross-striated?
- 2) Better defining of the two muscle types when they are first introduced in the abstract would help. I don't think most researchers think of muscle types as cross-striated versus not cross-striated.
- 3) Similarly, a better description of tubular muscles the first time it's mentioned and defining intercalation in the context here would be helpful to increase the reader's immediate understanding.

REVIEWER COMMENTS

Reviewer #1 (Remarks to the Author):

This manuscript focuses on the relationship between myofibril architecture and mitochondrial structure in the Drosophila muscle system. The data are quite interesting in that changing mitochondrial morphology in the IFM yields structural changes in the myofibrils that apparently arise from the production of non-IFM isoforms of contractile proteins. The changes cause the fibrillar nature of the muscle to be transformed toward a tubular muscle type, such as leg muscle, with adjacent myofibrillar elements yielding a more dramatic cross-striation pattern along with centrally located mitochondria, rather than the intercalated pattern typical of IFM. The authors have developed new methodologies for assessing mitochondrial morphology and present excellent images to back up their conclusions. The movies are particularly striking. Statistical work appears complete and data are presented in detail in a supplementary table.

Specific results show that genetically reducing mitochondria size prevents normal IFM function, but does not affect the level of cross striation. In contrast, enhancing mitochondrial fusion changes mitochondrial location and level of cross striation toward the tubular muscle phenotype. This change occurs at the transcriptional level for at least three contractile protein isoforms and mitochondrial fusion needs to occur during myofibrillogenesis for the effect to be seen. Developmental studies are nicely done to show how mitochondrial intercalation occurs and how it is compromised in Marf1 IFM. Overall, the manuscript highlights a new area of investigation, the effect of mitochondrial morphology on myofibril organization, which will be of strong interest to individuals studying muscle structure, function and disease.

We thank the reviewer for appreciating the high quality, novelty and significant interest of our findings for the muscle community.

There are a number of areas that need to be improved:

1. When mentioning mechanical tension of myofibrils as causative of mitochondrial morphology, does this correspond to tension induced by actin-myosin interaction? Does removing this interaction genetically or with drugs affect mitochondrial morphology? Does removing thick or thin filaments affect mitochondrial morphology? The latter question likely can be answered by reference to the literature.

We thank the reviewer for asking to go deeper into this interesting mechanism. Our live dissection method had already collected evidence that it is indeed the myofibrils that produce the tension squeezing mitochondria into the ellipsoid shape (Fig. 3a-e). According to the reviewer's suggestion we have now directly tested a role of actin-myosin interaction by genetically removing myosin heavy chain in the flight muscles with the striking result that mitochondria are losing their elongated morphology and form spheres (new Fig. 3f,g). Similar observations have indeed been made in the past using electron microscopy of the myosin mutants. We thank the reviewer for this suggestion and have included the reference.

2. The introduction to Spalt knockdown in IFM on pages 6-7 does not adequately reference that this work was previously published, i.e., it reads as if it is a new finding. Perhaps the mitochondrial morphology was more closely examined here? In any case, reference 17 should be more specifically cited for the original discovery.

We formulate now more clearly that Spalt as a flight muscle selector gene was published previously. Mitochondrial morphology had not been investigated in this older study.

3. What about genes downstream of Spalt, since Spalt levels are normal? For instance, is Arret/Bruno splicing factor regulated? This seems an important question to address regarding mechanism.

Indeed, another interesting point for which we thank the reviewer. We had shown previously that Aret/Bruno requires Spalt for its expression and is responsible for a large part of the flight muscle-specific alternative splicing. We have now investigated Bruno expression at 24 h APF in *Mef2::Marf-1* flight muscles and find it is expressed normally (new Fig. 5q-r). This supports our interpretation that flight muscle fate is set up normally in *Mef2::Marf-1* flight muscles with the key transcriptional regulator (Spalt) and splice regulator (Bruno) being expressed.

How the change in mitochondria morphology is affecting the transcription (Fln-GFP, Act88F-GFP) and also splicing (Kettin-GFP) mechanistically downstream of Spalt and Bruno is now referred to in the discussion and will be a topic of future investigations.

4. On page 10: "over-expression of Marf with Mef2-GAL4 results in fewer flight muscles, likely due to muscle atrophy during development" Do the authors have any data on this issue to bolster their contention? Other possibilities exist, such as failure to form fibers or fusion of existing fibers.

We thank the reviewer for pointing out the concern that over-expression of Marf may affect myoblast fusion early during development, rather than resulting in muscle atrophy at a later stage. We have now addressed this concern and quantified nuclei numbers in *Mef2::Marf-1* flight muscles compared to wild type at 24 h APF and find comparable numbers (new Supplementary Fig. 4). This together with the presence of the 6 DLM fibers at 24 h APF (see new Supplementary Fig 4a,b) and the normal expression of Spalt and Bruno strongly suggests that a loss of *Mef2::Marf-1* flight muscle fibers (muscle atrophy) occurs after 30 h APF and is likely caused by the mitochondrial defects.

5. Do the authors know what actin isoform is expressed when Marf is over expressed in IFM? This would be interesting to find out. Specific muscle form? Cytoplasmic form?

These are likely Act79B or Act57B, which are the other muscle actins, possibly also Act87E. However, to our knowledge there are no specific antibodies or other protein fusion reagents under endogenous control, such as fosmids (as we have for Act88F-GFP), available. Hence, to date we do not know the specific identify of the other actin isoform gained upon *Marf* over-expression.

6. Does kettin in the IFM arise from alternative splicing or from using a different transcriptional start site, i.e., is the change due to splicing or transcription? This is related to the question as to whether only transcriptional changes (actin, flightin) occur or whether alternative splicing is altered as well.

This is another excellent point of the reviewer. As per the above suggestion (point 3) we have tested for expression of the splice regulator Bruno and it is expressed normally at 24 h APF. However, the expression of the short SIs isoform Kettin is indeed regulated by an alternative splicing mechanism that is normally specific to leg muscles (see Spletter et al. EMBOR 2015). This suggests that despite the normal expression of Bruno at 24 h APF the splicing program is also partially changed to leg muscles, likely at later stages of myofibrillogenesis. We have added this aspect to the discussion part.

7. Using the Actin88F promoter to express Gal4 and marf-1 is of some concern, since Actin88F production is turned down dramatically when marf-1 is over-expressed. Thus, one would think that Gal4 expression would be reduced as well. Does this explain the reduced phenotype compared to Mef2-Gal4?

This is exactly the rationale that we used when we designed the experiment. The Act88F promoter is normally very strong, but only active from 24 h APF onwards. Expression of Marf with this promoter will feedback and reduce its strength. Hence, over-expression of Marf with Act88F-GAL4 starts only at about 24 h and will be less strong compared to continuous expression with Mef2-GAL4. Hence, the phenotype is weaker.

8. Were DVMs and DLMs studied? Do they show any developmental differences in mitochondrial localization? Do the mitochondrial fusion and fission genes differentially affect them?

Both flight muscle types are very similar and appear equally affected. However, we have focused our studies mainly on the dorso-longitudinal flight muscles (DLMs) and have added a sentence to the text to make this clearer. See also 'minor point 11'.

9. Can the authors be more specific about the leg muscles examined within the Results section? Which leg? Which type of leg muscle?

We generally analyze leg muscles from the second thorax segment (second leg pair) and use the most proximal leg segment, the coxa, for most of the images. In the new leg muscle serial block-face electron microscopy acquisition we now included a low magnification image detailing the region and precise leg muscle used in the coxa muscle of T2. To the best of our abilities, we believe the leg muscles examined via light microscopy are very similar ones. We have included this information in the text (page 9).

Minor Concerns:

1. On page 8: "To segment the complex shapes of leg muscle mitochondria we used the mitochondrial matrix mito-GFP marker expressed with Mef2-GAL4 (Fig. 2j,k, Supplementary Movie 3)." However, panel 2j refers to IFM.

We thank the reviewer for spotting this error in the text. We have now carefully relabeled the panels and text, as we updated this figure to include a new data set of serial block face-EM for the leg muscle.

2. On page 9: "Strikingly, severing myofibrils results in a dramatic rounding up of all neighbouring mitochondria into spheres (Fig. 3c,e). This transition was observed with both mitochondrial markers, MOMGFP (Fig. 3c) as well as with mito-GFP (Fig. 3e)." References to Fig. 3e should actually reference Fig. 3e'.

We thank the reviewer for the remark. We have corrected 3e for 3e' now in page 10.

3. On page 10: "which adopt a spherical instead of an ellipsoid shape (Fig. 4d,e,k)" should reference 4d' and 4e'

We have updated the text accordingly.

4. Fig. legend 1: "Dotted lines in cross sections in (c'') and (h'') represent the regions measured." Should this be c''' ?

As we do not label the cross sections of a panel with yet another label, we have altered the legend to address this as follows: 'Dotted lines on the yz cross sections of (c'') and (h'') represent the regions measured'. We believe this is clearer and thank the reviewer for the remark.

5. Fig. 2 panel h is not labeled.

Thanks. We have now corrected all the labels in this figure, since we re-arranged the panels as noted above.

6. Fig. 2 legend: "intimated contact": I presume the authors mean "intimate contact" Also, panels n, o, o' and o'' are not correctly cited in the legend.

We thank the reviewer for catching the typo, it has been corrected. We have also corrected for the panel labeling, since we re-arranged this figure as mentioned above. Panels n, o, o' and o'' have now been moved to Supplementary Figure 2g, h, h' and h''.

7. Fig. 5 legend: "GFP fluorescence levels are represented via a heat map (fire LUT, white is higher)". I am not seeing a heat map and I don't know what the notation in parentheses denotes.

We have corrected the sentence in the legend that now reads: 'GFP fluorescence levels are represented via a pixel intensity scale (white represents higher intensity)'.

8. Fig. 6 legend (and paper text) regarding Act88F driver: "largely cross-striated myofibrils (k)". This description appears to be an over-statement.

In the manuscript we wrote 'partially cross-striated myofibrils' and now the Figure 6 legend has been corrected to read the same.

9. Figures 7 and 8 appear somewhat redundant. Perhaps they can be combined?

We understand the remark of the reviewer. However, we consider the point of the mitochondria intercalation during development to be very important and prefer to highlight it using both GAL4 driver lines given their different temporal dynamics. The later onset and somewhat milder phenotype of the Act88F-GAL4 driven Marf expression allowed us to also reliably study the 48 h APF time point, at which the intercalation defect is most dramatically highlighted. This is best illustrated in larger images which needs two figures. As Nature Communications does not publish in print, space limitations should not be too strict.

10. In both the introduction and discussion, more effort should be made to differentiate vertebrate and fly data. The reader may not appreciate that there are differences between the systems. For instance: “more elongated mitochondria present in slow fiber types” is presumably referring to vertebrate muscle only; “How these switches are controlled is not fully understood, but it is conceivable that changes in mitochondrial metabolism may contribute [talking about mammals]. Alternatively, as manipulating mitochondrial dynamics affects myofibril alignment, this change in the biomechanical properties of the myofibrils may impact the transcriptional status of the muscle fiber [talking about flies and possibly applying this to mammals].”

We thank the reviewer for this suggestion. In the revised version we have now made it clearer when we refer to fly or mammalian muscles and which data came from which systems.

11. The use of “flight muscles” instead of “indirect flight muscles” or IFMs could prove confusing to the reader, as direct flight muscles are not studied here. Being more precise throughout or at least indicating the shorthand labeling of IFMs as flight muscle at the beginning of the manuscript would be important.

We have clarified this point now in the beginning of the results section and also made it clear which indirect flight muscles we are analyzing: ‘Flight muscles consist of dorso-ventral (DVMs) and dorso-longitudinal muscles (DLMs) . As both show a very similar morphology, we focus on the DLMs and for simplicity call them flight muscles in the remainder of the manuscript.’

We hope this is clear now and still avoids the use of specialized terminology throughout the manuscript.

12. The term “cross-striated muscles” can be confusing, since all insect muscle is considered striated. While the leg muscle pattern is dramatic as opposed to the poorly aligned fibrillar nature of IFM, the authors use a striation index that shows a positive value in IFM. Hence this issue needs to be more carefully explained.

With ‘cross-striation’ we exactly want to highlight this difference of aligned leg muscles compared to flight muscles, which ‘only’ show striated myofibrils. Our cross-striation index (Csi) aims to quantify this difference. Csi will always be positive and score ‘1’ for a perfectly aligned muscle. Lower values show a degree of misaligned fibrils, as the case for fibrillar flight muscles. 0.5 is the lowest possible value. We have now modified our diagram in Supplementary Figure 1 to better explain the range of values expected in each situation.

Reviewer #2 (Remarks to the Author):

The present paper addressed an interesting aspect of muscle morphogenesis related to the cross-talk between mitochondria and myofibrils. Initially, the authors characterized the morphology of mitochondria and myofibrils in leg and flight muscles and showed that the leg muscles displayed a cross-striated myofibrils while flight muscle a fibrillar morphology. This myofibrillar morphology with densely packed ellipsoid-like mitochondria at the periphery was dependent by the transcription factor Spalt. Beautiful 3D reconstruction images of mitochondrial networks underline the differences of mitochondrial shape between flight and leg muscles. Due to these observations and to the finding that a drop in mechanical tension results in mitochondrial shape change from ellipsoid to spheres losing the contact site with myofibrils, they postulated that mitochondrial dynamics may contribute to myofibrils morphogenesis. Gain and loss of function experiments for the fusion and fission mitochondrial proteins confirm that the shape of the network is important for the flight muscle function and for the mitochondrial intercalation between myofibrils. The work is well described and the images of mitochondrial network of leg and flight muscles are beautiful. The take home message is of interest for the community and further underline the critical role of mitochondria as hub of signaling pathways that retrogradely modulate genes transcription. However, some aspects are merely descriptive and need more insights. The following points should be considered.

We thank the reviewer for pointing out the high quality of our figures and animations, as well as the importance and impact of our findings for the community.

Point1. The idea that mitochondrial network may drive the morphogenesis of myofibrils is interesting. However, the data that Marf knockdown or DRP1 overexpression does not change the fibrillar morphology (despite mitochondria are now rounded that is against a mechanical tension effect on mitochondrial shape) while Marf overexpression does, needs a rationale for the correct interpretation of these experiments that is to monitor Marf and Drp1 expression in leg and flight muscle before and after 24 h APF.

Our line of argumentation is that small mitochondria or less connected mitochondrial networks can intercalate between the assembling myofibrils at 30 h APF in flight muscles. This is the case in wild type flight muscles and in *Marf* knock-down or *Drp1* over-expression flight muscles, in which mitochondria are small, but not in *Marf* over-expression or *Drp1* knock-down flight muscles, in which mitochondria cluster together. We have now added new electron microscopy data to also show at ultrastructural level that mitochondria are clustered in *Mef2::Marf-1* flight muscles compared to wild type at 24 h APF. This prevents intercalation (new Fig. 7i-l).

We have also added expression time course data that illustrate the levels of *Marf* and *Drp1* RNA in wild type flight muscles. This shows that *Marf* levels in wild type only increase after 30 h APF and *Drp1* levels drop after 48 h APF. This presumably is needed to allow mitochondrial growth when the myofibrils mature after 48 h APF. But at myofibril assembly stage the level of the fusion promoting *Marf* is low, whereas the level of the fission promoting *Drp1* is relatively high (new Supplementary Fig. 6). This allows intercalation.

Point2. Images of mitochondrial-myofibrils network of MARF overexpressing flight muscles does not show the same quality of the initial figures. 3D reconstructions (including EM) should be also produced for these experiments (Fig 6-7-8).

We thank the reviewer for this important comment. We have now further improved our imaging for all genotypes investigated. We now provide high-resolution airyscan deep stack acquisitions for all panels documenting in detail mitochondria morphogenesis during flight muscle development (Fig. 7 and 8). We further have generated 3D reconstructions, with stills included in Fig. 7 and 8 as well as high resolution animations included in the Supplementary Movies 8 and 9. This documents the interaction between myofibrils and mitochondria during development.

Segmentation of individual adult flight muscle mitochondria was possible due to their larger size, whereas during early stages individual mitochondria are too small to be reliably separated. Our highly resolved confocal microscopy imaging reliably documents how mitochondria intercalate between the assembling myofibrils between 24h and 48h APF, whereas they cluster upon Marf over-expression blocking their intercalation.

To reliably document individual mitochondrial shape we have performed electron microscopy at the key stage of 24 h APF in wild type compared to *Mef2::Marf-1* flight muscles. These data document the clustering of the mitochondria compared to the individual elongated structure in control. These new data are now included in new Fig. 7i-l.

To document the leg muscle mitochondria morphology at the same detail as for the flight muscles we have now performed serial block face electron microscopy of an entire leg muscle and reconstructed the detailed morphology of individual mitochondria (included in Fig. 2 and new Supplementary Movie 6). These results further strengthen our conclusions based from light microscopy that leg muscle mitochondria do not systematically intercalate between myofibrils and are largely located above or below the myofibril layer, however send thin extensions towards the sarcomeric I-bands. This documents their complex shapes now with much higher confidence (new Fig 2i-q).

Point3. Mitochondrial network was not shown when dnDRP1 was expressed while it should (Suppl Fig 3h-l). Blocking fission does not mean that the network is identical to the one of Marf overexpression.

We thank the reviewer for this important comment. We have now visualized mitochondrial distribution as well as a quantification of the cross-striation phenotype in the dominant negative Drp1 flight muscles (new Supplementary Fig 3h-k). Data show that the dnDrp1 results in a very similar phenotype as Marf over-expression with clustered mitochondria and a high alignment of flight muscle myofibrils. This nicely confirms the main conclusion of our manuscript.

Point4. The atrophy observed in Marf1 and dnDRP1 overexpressing flies is problematic because per se it can interfere with myofibril morphogenesis. Is this atrophy due to increased protein breakdown?

It is difficult to know why exactly some flight muscle fibers undergo atrophy in *Mef2::Marf-1* flies. We did not investigate if this is due to changed protein turnover. However, atrophy is reduced in the milder *Act88F::Marf-1* condition, which essentially results in the same mitochondrial intercalation defect and a similar change in transcription and cross-striation.

Hence, we feel our conclusions that 'Myofibril and mitochondria morphogenesis are coordinated by a mechanical feedback mechanism in muscle' are solid.

Point5. Act88F, Flightin and Kettin expression should be monitored also in Act-88F-Marf-1 line.

We thank the reviewer for this very useful suggestion. Now we provide new data that show a very similar change of Act88F-GFP, Fln-GFP and Kettin-GFP expression in *Act88F::Marf-1* flight muscles as seen before in *Mef2::Marf-1* (new Supplementary Fig. 5).

Point6. It would be important to understand which step downstream Spalt is regulated by Mitochondrial dynamics

Of course the reviewer has an interesting point here. In the updated discussion we are providing several possible solutions how either a change in metabolic status or a direct change in the biomechanics of the assembling myofibrils may feedback on the transcriptional status downstream of Spalt. As this is the first paper finding a mechanical interaction between myofibrils and mitochondria, as well as a transcriptional feedback, we need to await further experiments to reveal the mechanistic detail how the transcriptional response is regulated molecularly.

Reviewer #3 (Remarks to the Author):

This is an outstanding paper investigating the synergistic development of muscle mitochondria and myofibrils. The paper combines sophisticated imagery with cutting edge computer aided quantitative analysis, the power of Drosophila genetics and mutants, and leverages the contrasting properties of two Drosophila muscle types, fibrillar versus cross-striated. The images and videos are not only very informative but entrancing and beautiful.

We are grateful to the reviewer for the very enthusiastic assessment of the quality and illustration of our study.

Some of the new and novel findings that significantly advance our understanding of muscle development include: In fibrillar muscle where each myofibril is surrounded by mitochondria, mechanical pressure from the myofibrils force mitochondria into an elongated shape that causes a high degree of contact between the two. This intimate contact presumably helps with delivery of ATP to myofibrils. Conversely, the elongated mitochondrial that are squeezed between the myofibrils, help prevent individual myofibrils from fusing together to form cross-striated myofibrils. One method by which this was shown was by genetically increasing fusion rate to prevent long intercalated myofibrils. This also helps regulate myofibril size. The authors show that cross-striated muscle forms, at least in part, due to higher fusion rate of mitochondria.

The paper includes excellent use of Drosophila genetic to show that altering mitochondria formation, particularly fusion, leads to converting non-striated muscle to a striated muscle morphology, i.e. by disrupting mitochondrial intercalation. This conversion included altering expression of isoform specific proteins of the myofibril, which raises the very interesting

question of how this feedback occurs.

The methods all appear to be sound and I did not see any problems. I particularly liked the mechanical disruption of tension on myofibrils and mutants that increase or decrease mitochondrial fusion.

Good controls are included such as altering Spalt expression levels, and appropriate use of flight tests to verify that the results are physiologically important. Interpretation also appears quite sound and backed by strong experimental results.

I think this paper will be of high interest to researchers in several fields including developmental biology, mitochondria biogenesis, muscle energetics and muscle mechanics. I predict it will be very well received.

We thank the reviewer for this very detailed summary of the key findings and for highlighting their quality and impact for the muscle and mitochondria communities.

Minor concerns:

1) In the discussion, the authors could consider how their findings with Drosophila relate to the different muscle types found in mammals that they mentioned in the introduction. The introduction mentions higher endurance muscles (which presumably includes slow-twitch fiber types?) have more elongated mitochondria. Do these types also have less fused mitochondria than fast twitch? The authors include a good discussion of heart muscle, but how about skeletal? What type of myofibrils do fast and slow twitch have? Cross-striated?

This is an interesting point. To our knowledge and from what we find in the literature, both, slow and fast twitch muscles do have cross-striated myofibrils. Still their mitochondria shapes differ significantly with more mitochondria in proximity to the myofibrils in oxidative (slow) fibers. If this difference in mitochondria shape is instructive for possible differences myofibril morphology between both types is interesting and would need to be explored in detail. It appears challenging with current technologies to combine the detailed mitochondrial morphology at the EM level with the exact knowledge of fiber-types in mammalian muscle (see Vincent et al. Cell Reports 2019), however it has been approached in one recent study (Bleck et al. Nat. Comm. 2018). As all mammalian muscles consist of mixed fiber types this is a challenge. Decisions of fiber types are often made on pure mitochondrial content instead of myosin isoform markers, which only work at the light microscopy level. In flies, we have 'pure' fiber types.

2) Better defining of the two muscle types when they are first introduced in the abstract would help. I don't think most researchers think of muscle types as cross-striated versus not cross-striated.

It is hard to provide a detailed introduction within the word limit of the abstract. However, we have improved the introduction to clarify the differences of the *Drosophila* muscle types in the manuscript.

3) Similarly, a better description of tubular muscles the first time it's mentioned and defining intercalation in the context here would be helpful to increase the reader's immediate understanding.

Thanks for this comment. We have updated our introduction to be clearer.

REVIEWERS' COMMENTS

Reviewer #1 (Remarks to the Author):

My comments indicating my high enthusiasm for this work are included in my first review. The authors did an outstanding job in responding to my concerns. The only one that remains is the comment in Fig 5 and Supplementary Fig 5 that "GFP fluorescence levels are represented via a pixel intensity scale (white represents higher intensity)". While the authors tried to address this concern in revising the sentence, I remain unclear how to interpret the scale by only knowing that white represents higher intensity. I am assuming there is a series of colors that indicate increasing intensity that culminates in white, but without knowing more details of the series, how can one interpret the fluorescence levels? I may be misunderstanding the methodology, but this might be the case for other readers.

Sanford I. Bernstein

Reviewer #2 (Remarks to the Author):

The authors addressed my comments. There is only one minor issue. On page 12 line 264 authors claimed "...likely due to atrophy during development". This is incorrect, atrophy occurs only when adult not growing cells shrink their volume due changes in protein turnover. The decreased size due to a development defect is not atrophy. Authors should change the sentence in "...likely due to a reduced growth"

Reviewer #3 (Remarks to the Author):

The authors have done a great job addressing my previous comments.

REVIEWERS' COMMENTS

Reviewer #1 (Remarks to the Author):

My comments indicating my high enthusiasm for this work are included in my first review. The authors did an outstanding job in responding to my concerns. The only one that remains is the comment in Fig 5 and Supplementary Fig 5 that "GFP fluorescence levels are represented via a pixel intensity scale (white represents higher intensity)". While the authors tried to address this concern in revising the sentence, I remain unclear how to interpret the scale by only knowing that white represents higher intensity. I am assuming there is a series of colors that indicate increasing intensity that culminates in white, but without knowing more details of the series, how can one interpret the fluorescence levels? I may be misunderstanding the methodology, but this might be the case for other readers.

Sanford I. Bernstein

We appreciate once more the strong support and enthusiastic words from the reviewer. We understand the concern about the unclear intensity scale legends in Figure 5 and Supplementary Fig 5. We have now added colour scale bars to both figures that clearly indicate the range of colours used and how they relate to relative intensities displayed. We also added more explanation to the figure legend making clear that these colours represent relative intensities. We believe this is now more easily comprehensible.

Reviewer #2 (Remarks to the Author):

The authors addressed my comments. There is only one minor issue. On page 12 line 264 authors claimed "...likely due to atrophy during development". This is incorrect, atrophy occurs only when adult not growing cells shrink their volume due changes in protein turnover. The decreased size due to a development defect is not atrophy. Authors should change the sentence in "...likely due to a reduced growth"

We thank the reviewer for the continued support of our manuscript. We agree with the remark and have changed the sentence to "... likely due to reduced growth during development".

Reviewer #3 (Remarks to the Author):

The authors have done a great job addressing my previous comments.

We thank the reviewer once again.